

# Development of Fully Interactive Hydrogen with Methane in UKESM1.0

Megan A. J. Brown[1], Nicola J. Warwick[1, 2], Nathan Luke Abraham[1, 2], Paul T. Griffiths[1, 2, 3], Steve T. Rumbold[4], Gerd A. Folberth[4], Fiona M. O'Connor[4, 5], and Alex T. Archibald[1, 2]

[1]Centre for Atmospheric Science, Department of Chemistry, University of Cambridge, Cambridge, CB2 1EW, UK
[2]National Centre for Atmospheric Science (NCAS), University of Cambridge, Cambridge, CB2 1EW, UK
[3]School of Chemistry, University of Bristol, Bristol, BS8 1TS, UK
[4]Met Office Hadley Centre, Met Office, Fitzroy Road, Exeter, Devon, EX1 3PB, UK
[5]Department of Mathematics & Statistics, Global Systems Institute, University of Exeter, Exeter, EX4 4QF

**Correspondence:** Megan A. J. Brown (majb4@cam.ac.uk) and Alex T. Archibald (ata27@cam.ac.uk)

**Abstract.** Hydrogen is a potential candidate for an alternate energy source and carrier. As usage of hydrogen in industry rises, leakages into the atmosphere may occur, causing an increase in the global atmospheric hydrogen concentration. Hydrogen is an indirect greenhouse gas, known to increase methane, stratospheric water vapour, and tropospheric ozone. Methane and hydrogen are closely coupled, with the main atmospheric destructive pathway of both species being via reaction with the

hydroxyl radical (OH). Currently, most earth system models (ESMs) simulate hydrogen or methane with a prescribed lower boundary condition, which suppresses chemical feedbacks at the surface. In this work, we implement hydrogen emissions and a hydrogen soil uptake scheme into an ESM with free-running methane to demonstrate the capability of a fully interactive hydrogen and methane emissions-driven ESM. We show that the model is able to capture long term trends and seasonal cycles of both species when compared to observations, and find that the inclusion of both fluxes does not impact other chemical

species in the model, such as tropospheric ozone. We show that the model can be used under pre-industrial conditions and with a hydrogen pulse experiment. The ESM with fully coupled hydrogen and methane chemistry has great potential to be used in future scenarios and to estimate a more accurate global warming potential of hydrogen.

## 1 Introduction

Hydrogen is an alternate energy carrier to fossil fuels (Hydrogen Council, 2020). Transport of hydrogen is prone to leakage,
which may result in an increase of its atmospheric abundance. While not a greenhouse gas itself, hydrogen can indirectly contribute to greenhouse gas levels by extending methane lifetime, and increasing stratospheric water vapour and tropospheric ozone (e.g. Warwick et al., 2004, 2023; Tromp et al., 2003; Derwent et al., 2006). The main atmospheric chemical sink for hydrogen is via reaction with the hydroxyl radical (OH). Previous studies have found that OH is a competing resource between methane and hydrogen, as methane also reacts with OH as its main sink. Increasing hydrogen concentration therefore causes
an extension in methane lifetime, as OH is depleted by reaction with hydrogen (Warwick et al., 2023). Earth system models (ESMs) currently do not account for both hydrogen and methane fluxes at the surface, and often use a lower boundary condition





to prescribe the concentration of either $CH_4$ or $H_2$ or both (e.g. Sand et al., 2023; Paulot et al., 2021; Warwick et al., 2023; Bryant et al., 2024).

OH is a highly reactive radical, and there is much uncertainty for its reactivity and abundance (Yang et al., 2024; Chen et al., 2024). Given that hydrogen and methane are closely related and impact one another, coupling hydrogen and methane in an ESM will give a better understanding of the methane feedback factor from the impact of hydrogen. This work implements a hydrogen emissions-driven capability with a deposition scheme alongside the methane flux scheme from Folberth et al. (2022) to create a fully interactive hydrogen and methane model. We analyse the impacts of an interactive hydrogen scheme on the effects of methane, as well as comparing the simulated hydrogen and methane to observations. We then run the fully interactive ESM under pre-industrial conditions and also with a pulse of hydrogen in a present-day scenario to assess the feedback between these two species.

## 2 Methods

### 2.1 Model Configuration

The model used in this work is version 1.0 of the UK's Earth System Model (UKESM1.0), using the Unified Model (UM) at version 12.0. The model uses an atmosphere-only configuration (UKESM1.0-AMIP), and is set with 85 vertical levels and on a $1.875° \times 1.25°$ degrees (longitude-latitude) grid. The chemistry scheme implemented in UKESM1.0 is described and evaluated in Archibald et al. (2020). 'Nudged' simulations are set up as described in Telford et al. (2008), and use wind speeds ($u$ and $v$) and air temperature from ERA5 data. The control simulation has been run from 1982 to 2013 with both the hydrogen (time-invariant) and methane (time-varying) configured with a Lower Boundary Condition (LBC) which a uniform spatial distribution.

A total of four nudged simulations have been performed; a control, two runs with either $H_2$ flux or $CH_4$ flux turned on, and one with both fluxes on. Table 1 shows a full list of the simulations run and a description of each. The Sanderson simulation uses methane LBC with the hydrogen deposition scheme described in Sanderson et al. (2003). As hydrogen deposition has a very limited interannual variability, the Sanderson simulation was only run for one year to compare to the $H_2$-flux simulation. In addition to the nudged simulations, three timeslices were also performed. One is under pre-industrial (PI) conditions (year 1850), while the other two are present day conditions (year 2020). One of the present day simulations acts as a control, while the other is a hydrogen pulse experiment. Timeslices for present day ($H_2$; $CH_4$-PD) and pre-industrial ($H_2$; $CH_4$-PI) with both interactive hydrogen and $CH_4$ are spun up for 30 years, with the former using emissions from the Coupled Model Intercomparison Phase 6 project, using the Shared Socioeconomic Pathway 3 with a radiative forcing of 7.0 $Wm^{-2}$ at the end of the 21st century i.e., SSP3-7.0 (Rao et al., 2017; Riahi et al., 2017). The $H_2$; $CH_4$-Pulse experiment was conducted by setting the hydrogen concentration at all levels to 25% higher than the global-averaged surface concentration in the $H_2$; $CH_4$-PD simulation (530 ppbv). The hydrogen concentration was set to 45.7 ppb (662.5 ppbv) throughout the model, and used the same starting conditions as the spun up $H_2$; $CH_4$-PD simulation.





**Table 1.** List of simulations. Present day and pre-industrial simulations have been spun up for 30 years. LBC = Lower Boundary Condition, Flux = interactive tracer with emissions and uptake. PD = Present day, PI = Pre-industrial

| Name | Abbreviation | Description |
| --- | --- | --- |
| $H_2$ LBC; $CH_4$ LBC Transient | Control | Nudged from 1982. |
| $H_2$ Flux; $CH_4$ LBC Transient | $H_2$-flux | Nudged from $1982 - 2013$ with $H_2$ emissions |
| $H_2$ LBC; $CH_4$ Flux Transient | $CH_4$-flux | Nudged from $1982 - 2013$ with $CH_4$ biogenic, biomass burning and anthropogenic emissions + $CH_4$ flux adjustment (Folberth et al., 2022) |
| $H_2$ Flux; $CH_4$ Flux Transient | $H_2$; $CH_4$-flux | Nudged from $1982 - 2013$ with both $H_2$ and all $CH_4$ emissions |
| $H_2$ Flux; $CH_4$ LBC Transient | Sanderson | Nudged with Sanderson hydrogen deposition scheme from $1982 - 1983$ |
| $H_2$ Flux; $CH_4$ Flux PD Timeslice | $H_2$; $CH_4$-PD | 2020 Present day timeslice. SSP370 $H_2$ emissions for 2020. $CH_4$ emissions are same as Folberth et al. (2022) |
| $H_2$ Flux; $CH_4$ Flux Pulse Timeslice | $H_2$; $CH_4$-Pulse | 2020 Present day timeslice with hydrogen pulse. Same emissions as PD timeslice |
| $H_2$ Flux; $CH_4$ Flux PI Timeslice | $H_2$; $CH_4$-PI | Pre-industrial timeslice from 1850. $H_2$ and $CH_4$ emissions with PI $CH_4$ flux adjustment |

## 2.2 Hydrogen Emissions

55 For the sources and quantities of primary hydrogen input to the model, we follow the method of Paulot et al. (2021). Hydrogen is released into the atmosphere from biomass burning and anthropogenic fossil fuel combustion. In addition, natural hydrogen sources associated with marine and terrestrial nitrogen fixation are also included. A summary of the emissions used for the nudged simulations are shown in Figure A2.

As in Paulot et al. (2021), we use carbon monoxide (CO) emissions as a proxy for the evolution of hydrogen emissions
60 trend. To obtain the amount of hydrogen emitted at any given time, we scale the corresponding CO emissions using a source specific ratio. The ratio of CO:$H_2$ for each category (anthropogenic, biomass burning, oceanic and terrestrial) is derived from the period $1995 - 2014$, where we have a known estimate of hydrogen emission (Table 1; Paulot et al., 2021) and can calculate the average CO emission over that time from emissions data.

The CO emissions used for the historical proxy are those created for UKESM1.0 for CMIP6 and detailed in Sellar et al.
65 (2020). For anthropogenic and biomass burning emissions we use a historical timeseries ($1850 - 2014$). Anthropogenic emissions are taken from the Community Emissions Data System (CEDS; Hoesly et al., 2018). For biomass burning we use those recommended for CMIP6 emissions (van Marle et al., 2017).




A seasonally varying climatology is used for natural emissions. Oceanic emissions use a monthly varying annual cycle for the year 1990 from the POET (Granier et al., 2005) inventory and apply it repeatedly. Terrestrial biosphere emissions are taken from the MACCity-MEGAN emissions inventory (Sindelarova et al., 2014), and a monthly varying annual cycle is taken from a mean of the years $2001-2010$. All $H_2$ emissions are regridded from their native horizontal resolution to the N96 model grid ($1.875° \times 1.25°$ longitude–latitude, approximately $135\,km$ resolution) conserving mass. The resultant hydrogen emission for each source (anthropogenic, biomass burning, oceanic, and terrestrial) follows the spatial pattern of the equivalent CO source, but with values rescaled to give the global emission total appropriate for hydrogen.

The year-2020 anthropogenic and biomass burning hydrogen emissions use the same source specific ratios of $CO:H_2$ as described above, and have been applied to year-2020 CO emissions from CMIP6 SSP3-7.0 (Gidden et al., 2019). The monthly varying climatologies for natural oceanic and terrestrial biosphere hydrogen emissions are invariant in time and, thus, are as described above.

## 2.3 $CH_4$ Flux Adjustment

In order to bring the global surface methane concentration into agreement with observations, a residual methane surface exchange flux, or flux adjustment is included in all simulations, as per Folberth et al. (2022). Methane emissions used in the $CH_4$ flux experiments include biomass burning, anthropogenic, and biogenic emissions. Three different sets of methane flux adjustments are used for the PI ($H_2$; $CH_4$-PI) timeslice, PD ($H_2$; $CH_4$-PD) timeslice and nudged simulations ($CH_4$-flux and $H_2$; $CH_4$-flux). The flux adjustments for the PI and PD timeslices are those described in Folberth et al. (2022). For the nudged simulations, a larger flux adjustment was required relative to the present day timeslice to bring modelled $CH_4$ in line with observations. Figure A1 shows the annually and globally averaged methane emissions, where the flux adjustment shown is used in the nudged simulation. Note that wetlands are excluded; these are calculated within the model and account for approximately $135\,Tg\,yr^{-1}$. This larger flux adjustment arises due to lower interactive wetlands emissions in the nudged simulations, compared to the present day timeslice ($135\,Tg\,yr^{-1}$ vs. $190\,Tg\,yr^{-1}$) which leads to an underestimate in modelled methane concentrations. This difference in global wetland emission is likely driven by differences in the global interactive wetland fraction, resulting in as which is one of the main driving factors of the interactive wetlands, is 0.0321 in both $CH_4$-flux nudged simulation. In comparison, the $H_2$; $CH_4$-PD wetland fraction at 0.0358, which is $11.5\%$ larger and could explain some of this discrepancy. In order to compensate for the low wetland emissions in the nudged simulations, the flux adjustment from the present day timeslice was increased to incorporate a global $20\%$ increase in methane emissions.

## 2.4 $H_2$ Soil Deposition Scheme

The model uses a two-layer hydrogen soil uptake scheme based on the work by Ehhalt and Rohrer (2013) and Paulot et al. (2021), and used in Brown et al. (2025). Processes in the first layer include the diffusion of hydrogen through the top layer of soil, while the second layer represents the loss from the microbial uptake of hydrogen. Two soil types, sand and loam, are included in the diffusion and microbial uptake parametisation. The sand fraction of each gridbox is used to classify the soil type, with the remaining fractions (loam and clay) classified as loam. This is likely to lead a slight overestimate in soil uptake





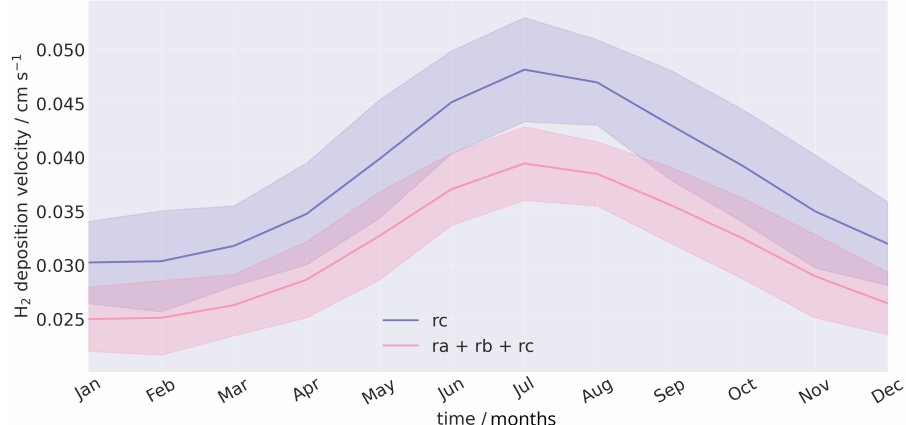

**Figure 1.** Monthly, land averaged hydrogen uptake for simulation $H_2$-flux ($H_2$ Flux; $CH_4$ LBC) between 1982-2013 with all resistances (pink) and with only hydrogen uptake (purple). Shaded areas indicate two standard deviations.

in areas of high clay content, as hydrogen uptake in clays is known to be lower than loam (Bertagni et al., 2021; Cowan et al., 2024). Full details of the deposition scheme are given in Section A1.

Further to the soil resistance (rc), the model also includes aerodynamic (ra) and laminar (rb) resistances (Marrero and Mason, 1972; Hicks et al., 1987). Including these resistances results in the global hydrogen uptake to decrease by $0.007\,\mathrm{cm\,s^{-1}}$ (average

$17.6\%$ decrease). The averaged land deposition with all resistances is $0.0314\,\mathrm{cm\,s^{-1}}$, while with just the uptake resistance the deposition is $0.0380\,\mathrm{cm\,s^{-1}}$. Figure 1 shows the monthly, land-averaged hydrogen uptake with (pink) and without (purple) the additional resistances. The largest changes due to aerodynamic and laminar resistances occur over South America and Africa (excluding the desert), where deposition velocities can have a temporally averaged maximum difference up to $0.14\,\mathrm{cm\,s^{-1}}$.

An offline version of this model used in Brown et al. (2025) implemented a soil moisture cap for areas with a high soil

moisture content (SMC), and a lower limit for uptake occurring in deserts. Both of these features are used in this model. If the volumetric SMC falls below the thresholds given by Equations A9 and A12, the values for $f_s(\Theta_a)$ or $f_l(\Theta_a)$, which govern the microbial uptake, are set to one third of the minimum values under those temperature and soil porosity conditions, reflective of the findings in Jordaan et al. (2020). For desert environments where the volumetric SMC is often below the threshold, this prevents the hydrogen deposition from immediately being set to zero, and allowing for uptake to occur in desert environments.

At locations of high volumetric SMC, hydrogen cannot diffuse through the top layer of soil and hydrogen deposition is very low. In UKESM1.0, the ratio of volumetric SMC to total porosity ($\frac{\text{volumetric SMC}}{\text{soil porosity}}$) is very high at northern latitudes with an annual average of $> 0.85$. Observed volumetric SMC during summer in these regions is between $0.15 - 0.3\,\mathrm{m^3\ m^{-3}}$ (Dorigo et al., 2023), which, if using the soil porosity from UKESM1.0, produces $\frac{\text{volumetric SMC}}{\text{soil porosity}} \leq 60\%$. This high ratio from UKESM1.0 is likely to be inaccurate due to the soil porosity not capturing the large coverage of peatland at these latitudes. Soil

porosity is calculated using sand fraction and does not take into account soils such as peatlands, which have been found to have a high hydrogen uptake (Simmonds et al., 2011). In order to compensate for the high volumetric SMC to porosity ratio, the





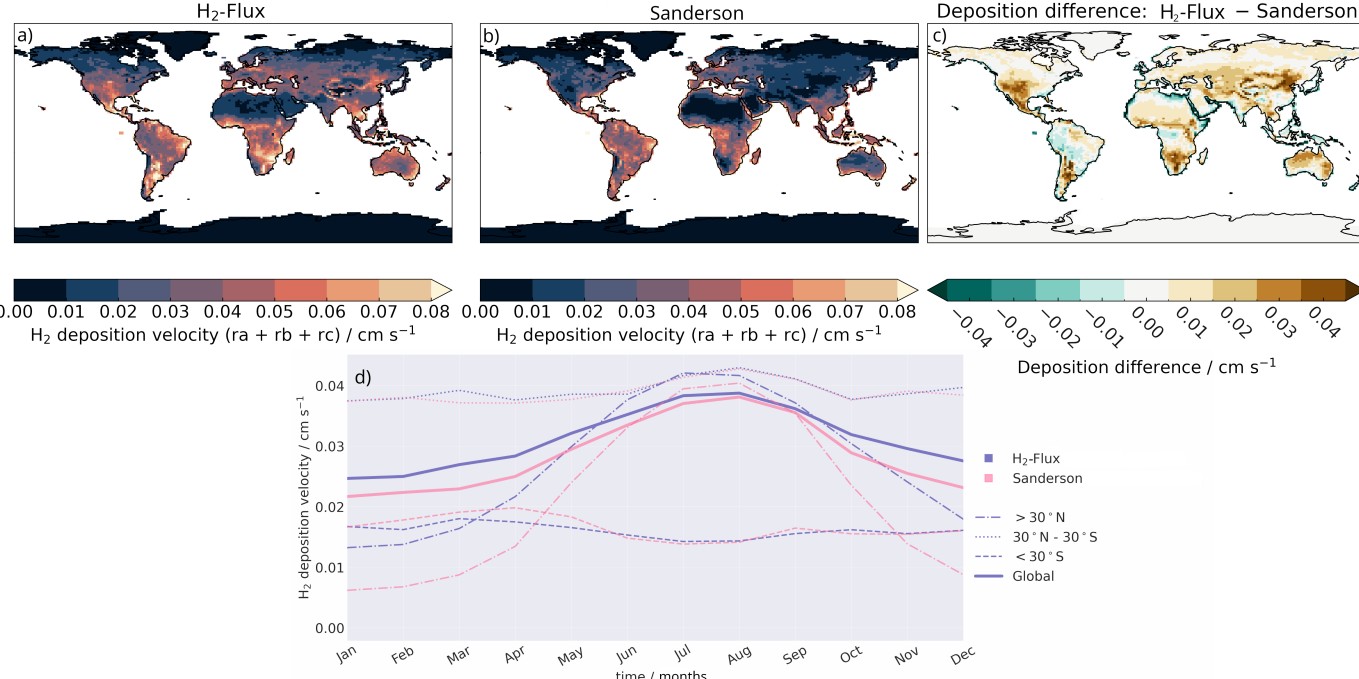

**Figure 2.** Annually averaged hydrogen deposition velocity in 1982 for a) $H_2$-flux using deposition scheme from Paulot et al. (2021) ($H_2$-Flux), b) Sanderson et al. (2003) deposition ($H_2$ Flux; $CH_4$ LBC), and c) difference between the two schemes (a − b). d) Land-averaged monthly mean hydrogen deposition of (purple) $H_2$-flux and (pink) Sanderson for 1982. Linestyles indicate $30°$ latitude bands into which data have been divided, with the thickest line showing the global-averaged deposition.

$\frac{\text{volumetric SMC}}{\text{soil porosity}}$ ratio in the hydrogen deposition scheme is capped at $70\%$. The volumetric SMC is adjusted so the maximum ratio does not exceed this value.

## 3    Evaluation of $H_2$ Deposition

The hydrogen deposition previously implemented in UKESM1.0 used the scheme from Sanderson et al. (2003). This deposition uptake was based on soil land type and scaled the deposition linearly or quadratically with soil moisture depending on the land type. While this produces global deposition velocities in line with other global hydrogen deposition values (see Sand et al., 2023), it does not include uptake in deserts, and is limited by requiring the land use type within a grid cell. The scheme adapted from Paulot et al. (2021) which is used in this work allows for an interactive hydrogen flux, which is dependent on soil properties and thus can be used for scenarios in which the land type is not known or prescribed (e.g. pre-industrial conditions, future climate simulations).

Figure 2 shows the comparison between the hydrogen deposition scheme based on Paulot et al. (2021) and the scheme from Sanderson et al. (2003). Both simulations have the same model configuration, using the same hydrogen emissions, 'nudged'



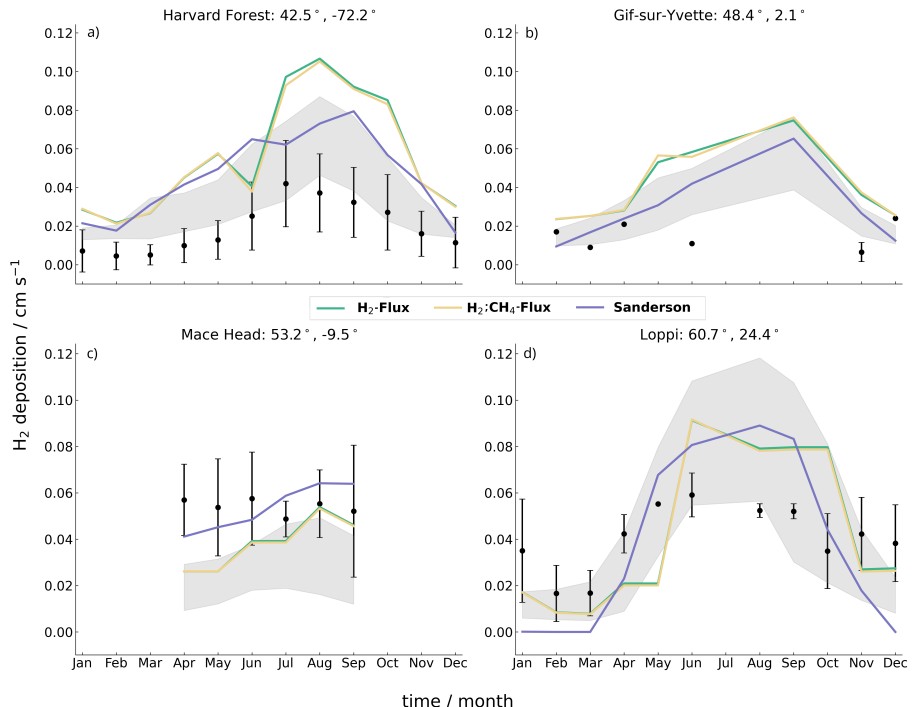

**Figure 3.** Observed hydrogen deposition velocities (points) compared with (green) $H_2$-flux, (yellow) $H_2$; $CH_4$-flux, and (purple) Sanderson simulated velocities for a) Harvard Forest ($2010-2012$; Meredith et al., 2017), b) Gif-sur-Yvette (2011; Belviso et al., 2013), c) Mace Head ($1995-2008$; Simmonds et al., 2011), and d) Loppi ($2004-2006$; Lallo et al., 2008). Error bars show one standard deviation across years. Shaded areas show the deposition range from CMIP6 models from Brown et al. (2025).

winds and air temperature from ERA5 data as per Telford et al. (2008). The $H_2$-flux simulation has a greater hydrogen depo-
sition in the northern hemisphere ($> 30°$ N) than in the Sanderson scheme, with deposition increase of $0.02-0.04\,\mathrm{cm\ s^{-1}}$ in
North America and Central Asia (Figure 2c). This is also seen by the solid lines in Figure 2d, which show the land-average
deposition above $30°$ N; the $H_2$-flux simulation (purple) is between $0.002-0.005\,\mathrm{cm\ s^{-1}}$ greater than the Sanderson et al.
(2003) (pink) simulation. Both simulations share a seasonal cycle which is dominant in northern latitudes, with the Sanderson
et al. (2003) simulation having a larger amplitude in seasonality. The $H_2$-flux simulation has an overall higher deposition ve-
locity (9.2% higher) throughout the year than that of the Sanderson et al. (2003) scheme (thick solid lines in Figure 2d), with
an annual, land-averaged mean of $0.0312\,\mathrm{cm\ s^{-1}}$ compared to $0.0285\,\mathrm{cm\ s^{-1}}$ respectively. This is due to uptake occurring in
the desert, as well as in the tropics and at high northern latitudes.

Figure 3 shows simulated hydrogen deposition against observations at four different sites, all of which are located in the
northern hemisphere. The land types of a) Harvard and d) Loppi are deciduous and boreal forests respectively, while b) Gir-
sur-Yvette and c) Mace Head are grassland/agriculture and peatbogs respectively. The soil types of all sites are different, with
Harvard Forest as sandy loam, Loppi as mineral and peat soils, and Mace Head as peatland (Gif-sur-Yvette not given). $H_2$-flux



and $H_2$; $CH_4$-flux simulations were compared against the observations. Both simulations produce very similar hydrogen velocities, globally and at all individual sites, which is expected as there is little atmospheric influence on the hydrogen deposition. Shaded areas show the range of deposition from CMIP6 models from Brown et al. (2025), while the purple line shows the deposition from the Sanderson simulation, averaged over one year. Compared to observations and other simulations, the $H_2$-flux and $H_2$; $CH_4$-flux simulations tend to overestimate deposition velocity, with the exception of Mace Head.

At Mace Head, the hydrogen values are underpredicted in April-May and are below the average observed concentrations, although within the error for the remaining months (June to October). Gif-sur-Yvette lacks observations during the summer months and thus there is no defined peak to which the simulations can be compared directly. However, between November-April, deposition velocities are within $0.005\,\mathrm{cm\,s^{-1}}$ of the observations. In both Harvard Forest a) and Loppi b), deposition is overpredicted by $0.02-0.04\,\mathrm{cm\,s^{-1}}$, with the largest discrepancy for both sites occurring in the summer months. The CMIP6 range also overpredicts deposition at these sites, although not to the same extent (up to $0.02\,\mathrm{cm\,s^{-1}}$). Overestimates in the $H_2$-flux and $H_2$; $CH_4$-flux deposition velocities correspond with underestimates in hydrogen concentrations in the northern hemisphere. This is further discussed in in Section 4.1.

## 4  Results

### 4.1  Comparison to Atmospheric Observations

In order to verify the integrity of the model, the hydrogen and methane outputs are first compared to multiple years of observations. Figure 4 compares surface observations of a) hydrogen and b) methane with the simulated $H_2$; $CH_4$-flux values. Observations at two stations are shown; Mace Head in the northern hemisphere (MHD; green) and Cape Grim in the southern hemisphere (CGO; orange). The latitude and longitude of simulated values were matched by nearest neighbour to the closest location of the station sites. The simulated hydrogen concentrations for Mace Head are in good agreement with the observations. There is a small overprediction up to $15\,\mathrm{ppbv}$ between $1994-2000$, but after 2000 the modelled hydrogen is in line with the observations. The modelled hydrogen at Cape Grim is consistently overpredicted during the southern summer, where hydrogen can be up to $35\,\mathrm{ppbv}$ greater than observed values. The minimum hydrogen concentrations are captured during southern winter, suggesting that the seasonal cycle of hydrogen is too strong in the model. This may be due to too extreme changes in soil moisture between the winter and summer months.

Simulated methane is slightly underpredicted when compared with observations from Cape Grim. This is seen in Figure 4b from 1995 onwards, where there is a $50\,\mathrm{ppbv}$ low bias with respect to observations in the model. Similarly, in the northern hemisphere, methane at Mace Head is generally in good agreement with observations from 1997 onwards ($50\,\mathrm{ppbv}$ excess). Between $1992-1997$, there is a larger difference, which reaches up to $80\,\mathrm{ppbv}$. The model captures the overall trend and seasonal cycle of observations.

Further to these two sites, the simulated hydrogen is compared to observations at all NOAA stations (Pétron et al., 2024). Observed data have been monthly averaged to match the same time resolution as the modelled data. Simulated hydrogen has been matched to observations by nearest latitude and longitude, and time up to the year 2014. Hydrogen observations after the



**Figure 4.** Time series of Mace Head for (MHD: dark green) observations and (light green) $H_2$; $CH_4$-flux simulation, and Cape Grim (CGO: orange) observations and (yellow) $H_2$; $CH_4$-flux for a) hydrogen (1994-2013) and b) methane (1985-2013).



year 2014 are discarded as simulations were run for the historical time period used in CMIP6, where there was availability
of historical emissions. Figure 5a summarises the variance via normalised standard deviation (NSD) and correlation between
simulated and observed hydrogen, while Figure 5b shows the average surface hydrogen concentration for the $H_2$; $CH_4$-flux
simulation (on map) and observation sites (points).

In Figure 5a, sites in the northern hemisphere are blue, while southern sites are in red. Table A1 gives the numerical key for

each site. Values close to the point marked 'REF' (which is 1) show the standard deviation of the model and observation site
are very similar, implying they have a similar variation. The correlation conducted uses a standard Pearson's correlation test,
and only stations with more than 20 datapoints pre-2014 were used in the experiment. Values with a higher correlation are able
to capture the seasonal trend of observed hydrogen better. Anomalies with either too high a NSD or negative correlation are
excluded from the diagram and written below with the format $\frac{\text{normalised standard deviation}}{\text{correlation}}$.

In Figure 5a, a greater proportion of southern sites (dark red) have a high correlation ($> 0.8$), suggesting they capture the
seasonal trend. The NSD tends to be greater than 1, implying that there is too much variation in the modelled data and that,
while the seasonal cycle is captured well, the amplitude in the simulated data is larger than the observations. For northern sites,
there is a spread of how well the simulated hydrogen captures the variation from observations. At sites where the model has
too much variation (NSD $> 1$), the correlation tends to be greater (between $0.65 - 0.95$), similar to the southern sites. This is

similar to the findings from the observed deposition at Harvard Forest and Loppi (Figures 3a and 3b), where there is too much
deposition occurring during the summer.

Values with an NSD $< 1$ tend to have a lower correlation, suggesting the model does not capture the observed variation at
these sites. Overall, most sites have an NSD around 1 and a correlation between $0.6 - 0.9$, indicating the simulation captures
the seasonal spread and cycle.

Figure 5b shows the simulated surface hydrogen concentration between $2008 - 2013$ with the averaged hydrogen concentration from all sites given as points. The simulated data captures the magnitude of the hydrogen concentrations at nearly all
observation sites. The exception to this is the very high ($> 620\,\mathrm{ppbv}$) hydrogen concentration over China and India, which results from the combination of anthropogenic emissions and orography in the model (Hayman et al., 2014). Overall, the model
slightly overpredicts hydrogen in the southern hemisphere by an average of $13.2\,\mathrm{ppbv}$, while in the northern hemisphere there

is a small averaged underprediction of $2.1\,\mathrm{ppbv}$. This is likely because the soil sink is too strong during the northern summer
months and too much hydrogen is being deposited, as implied by Figures 3a and 3b. In general, the model is able to capture
the average magnitude of hydrogen concentration at most sites, and replicates similar correlations and variations to those in the
observations.

## 4.2  Effect on Atmospheric Composition

Ozone and OH are both trace gases sensitive to chemical fluctuations in the atmosphere, due to their high reactivity. Global
methane concentrations is relatively unchanged with interactive $CH_4$, and there is a small increase in the global hydrogen
concentration (from $500\,\mathrm{ppbv}$ in the control to $550\,\mathrm{ppbv}$ $H_2$-flux simulation for $2008 - 2013$) when adding in interactive $H_2$.
Thus, as expected when adding in both the hydrogen and methane flux, there is minimal impact on tropospheric ozone or





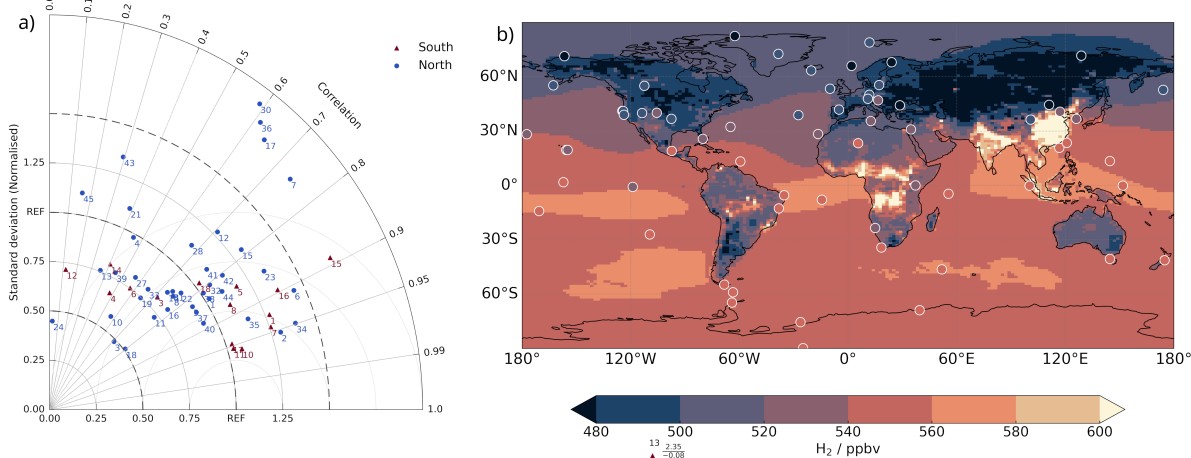

**Figure 5.** a) Taylor diagram showing Pearson's correlation and the standard deviation of modelled data ($H_2$; $CH_4$-flux) normalised to observations for all different sites. Red triangles show sites in the south, while blue triangles show sites in the north. Label correspond is sites and are given in Table A1. b) Map of averaged surface hydrogen concentration from $H_2$; $CH_4$-flux run overlaid with averaged hydrogen concentrations from all NOAA sites. All simulated data are averaged between $2008 - 2013$.

OH, as shown in Figure 6. There is a $30 - 60$ ppbv change in ozone between the control and $H_2$; $CH_4$-flux simulation in the troposphere ($< 0.02\%$ increase) and stratosphere ($0.06\%$ decrease). Similarly, there is less than $5 \times 10^4$ molecules cm$^{-3}$ increase in OH density at the surface with both hydrogen and methane flux ($< 0.02\%$ change).

The global surface average of hydrogen and methane is shown in Figure 7. In both the $H_2$; $CH_4$-flux and $H_2$-flux simulations, global surface hydrogen equilibrates to $545$ ppbv with an average $\pm 12$ ppbv seasonal cycle. In 1992 and 1998, there are sharp increases of hydrogen due to the increase of biomass burning in these years (Figure A2), and, in 2002, hydrogen equilibrates to between $540 - 550$ pbbv. The addition of the methane flux has little impact on hydrogen both globally averaged and across all latitudes (not shown). The control and $CH_4$-flux simulations both use a $H_2$ LBC of $500$ ppbv, which is constant across all times of year and latitude.

Figure 7b shows the surface global methane from the (grey) control and with methane flux with (red) $H_2$ LBC and (yellow) hydrogen flux implemented. Between $1996 - 2012$, methane in both simulations is slightly below the control by up to $30$ ppbv. The two simulations with methane flux are in good agreement with the LBC values (the latter of which is based on observations). They capture both the trend pre-1992 and the levelling off of methane abundance between $1998 - 2007$, as well as the increasing trend after 2007.

Methane decreases when the $H_2$ LBC is replaced by $H_2$ flux; methane is approximately $10$ ppbv lower. The decrease in methane in the $H_2$; $CH_4$-flux simulation occurs across all latitudes, with a greater decrease occurring in the northern hemisphere (not shown). One possible explanation could be due to an increase in OH from the increased hydrogen. The main atmospheric sink of methane is reaction via OH and thus dependent on OH availability;







**Figure 6.** Zonally averaged (top) OH concentration and (bottom) ozone concentration for the (left) control simulation, (middle) $H_2$; $CH_4$-flux simulation, and (right) the relative difference between the two ((Control $-$ $H_2$; $CH_4$-flux)/Control) averaged between $2008-2013$. Red (blue) shows an increase (decrease) from the control. Note the non-linearity of scales.

$$CH_4 + OH \rightarrow CH_3 + H_2O \tag{1}$$

To investigate further, we analyse the main chemical loss reactions (or fluxes) for hydrogen and methane. For methane, the dominant loss is $k[CH_4][OH]$ where species in brackets are their concentrations and k is the rate constant for the reaction. For
hydrogen, the loss terms are $k_1[H_2][O]$ and $k_2[H_2][OH]$, via the reactions;

$$H_2 + O \rightarrow OH + H \tag{2}$$

where the O can either be $O(^3P)$ or $O(^1D)$, and,





**Figure 7.** a) Globally averaged surface hydrogen from the (grey) control, (green) $H_2$-flux, and (yellow) $H_2$; $CH_4$-flux simulations from 1982 to 2013. Both the control and the $CH_4$-flux simulation have hydrogen set to 500 ppbv at the surface. b) Globally averaged surface methane from the (grey) control, (red) $CH_4$-flux, and (yellow) $H_2$; $CH_4$-flux simulations



$$H_2 + OH \rightarrow H_2O + H \tag{3}$$

with the latter being the dominant loss. Figure A3 shows the hydrogen and methane chemical loss; $H_2$ chemical loss increases

when $H_2$ flux is implemented into the model, while the $CH_4$ loss via OH decreases. This is expected, given that hydrogen concentration has increased and methane concentration has decreased. The fluxes are dominated by these changes in $[H_2]$ and $[CH_4]$ respectively, and the OH activity cannot be identified. By dividing through by each respective species, the loss reactions become independent of hydrogen and methane. The end column of Figure 8 shows the differences of (top) hydrogen lifetime via chemical loss and (bottom) $CH_4$ lifetime via OH loss or $\frac{1}{k[OH]}$ of methane loss. Blue (red) indicates an increase (decrease)

when $H_2$ flux is included, relative to the simulation with $H_2$ LBC.

In the northern hemisphere, there is an increase in the chemical loss rate both for $CH_4$ and $H_2$ (red in right column, Figure 8). This implies that the increase in $H_2$ loss rate is causing an increase in the $CH_4$ loss rate at these latitudes. The chemical loss of hydrogen is caused by Equations 2 and 3. The former of these reactions directly produces a OH molecule, while the latter could potentially generate more OH radicals by creating $HO_2$;

$$H + O_2 + M \rightarrow HO_2 + M \tag{4}$$

and reacting with $NO_X$:

$$HO_2 + NO \rightarrow NO_2 + \mathbf{OH} \tag{5}$$

$$NO_2 + h\nu \rightarrow NO + O \tag{6}$$


$$O + O_2 + M \rightarrow O_3 + M \tag{7}$$

Ozone can break down to indirectly produce more OH radicals:

$$O_3 + h\nu \rightarrow O(^1D) + O_2 \tag{8}$$

$$O(^1D) + H_2O \rightarrow \mathbf{2OH} \tag{9}$$

We find there is an increase in $k[HO_2][NO]$ at the surface which could support this mechanism, however it is not asymmetric between hemispheres and does not match up directly at the same latitudes (Figure A4).







**Figure 8.** Zonally averaged (top) $H_2$ chemical loss lifetime and (bottom) $CH_4$ lifetime via OH for (left) $CH_4$-flux simulation, (middle) $H_2$; $CH_4$-flux simulation, and (right) relative difference between the two ($CH_4$-flux − $H_2$; $CH_4$-flux) averaged between $2008 − 2013$. Red (blue) shows an increase (decrease) from the $CH_4$-flux. Note the non-linearity of scales. The stratosphere has been masked out to reduce noise.



**Table 2.** Hydrogen lifetime and budget of different simulations. Present day and pre-industrial timeslices are averaged over the last 5 years, nudged runs are averaged between $2003 - 2013$. Lifetime is given in years. Burden is in Tg, while chemical production, chemical loss, and emissions are given in Tg yr$^{-1}$.

| Name | $H_2$ total lifetime | $H_2$ soil lifetime | $H_2$ chemical lifetime | Burden | $H_2$ soil loss | $H_2$ atm loss | $H_2$ atm prod | Emissions |
|---|---|---|---|---|---|---|---|---|
| $H_2$-flux | 2.45 | 3.68 | 7.30 | 196.0 | 53.3 | 26.9 | 45.9 | 31.7 |
| $H_2$; $CH_4$-flux | 2.46 | 3.68 | 7.45 | 193.2 | 52.5 | 25.9 | 44.4 | 31.7 |
| $H_2$; $CH_4$-PI | 2.48 | 3.64 | 7.73 | 136.0 | 35.5 | 16.7 | 16.2 | 18.2 |
| $H_2$; $CH_4$-PD | 2.41 | 3.67 | 7.01 | 193.8 | 52.9 | 27.6 | 47.5 | 30.4 |
| Model-average from Sand et al. (2023) | 2.4 | 3.6 | 7.7 | 191 | 57 | 25 | 47 | 36 |
| Average from Ehhalt and Rohrer (2009) | 2.0 | - | - | 155[1] | 60 | 19 | 41 | 35 |

[1] This is tropospheric $H_2$ burden.

## 4.3 $H_2$ and $CH_4$ Budget

Both the hydrogen and methane lifetimes calculated in this work include the burden of the whole atmosphere. The hydrogen
lifetimes and burdens are summarised in Table 2, while the methane lifetime for the nudged simulations is summarised in Table 3.

Hydrogen lifetime is similar between all simulations, ranging between $2.41 - 2.48$ years. The soil lifetime remains constant, while the lifetime from chemical loss reactions fluctuates between $7.01 - 7.73$ years. Soil lifetime is expected to produce similar values between all runs, as the deposition is calculated through soil properties and is independent of hydrogen. In Sand
et al. (2023), soil lifetime was estimated from prescribed $H_2$ LBC values and both the soil and total model-averaged hydrogen lifetimes are similar to values in this work.

The changes between methane lifetime via OH loss are minimal, as shown in Table 3. Methane lifetime decreases when adding in methane flux (0.167 years; 2.1%), due to the slightly lower abundance of methane than the LBC condition, with is consistent with the findings from Folberth et al. (2022). When implementing $H_2$ flux into the ESM with a $CH_4$ LBC, there is a
small increase in methane lifetime. This is in agreement with the work conducted in Warwick et al. (2023), who ran a series of experiments with increasing $H_2$ LBC to assess the impact on methane. They found that when hydrogen abundance increased, the methane lifetime via OH reaction increased.





**Table 3.** Methane lifetime via OH loss in years from nudged simulations, averaged over 10 years between $2003 - 2013$ to 3sf.

| Name | $CH_4$ lifetime via OH |
|---|---|
| $H_2$ LBC; $CH_4$ LBC (Control) | 7.92 |
| $H_2$ flux; $CH_4$ LBC | 7.95 |
| $H_2$ LBC; $CH_4$ flux | 7.75 |
| $H_2$ flux; $CH_4$ flux | 7.75 |

When comparing the $CH_4$ flux simulations with and without $H_2$ flux, however, there is no significant difference in methane lifetime (both are 7.75 years to 2 decimal places). Further to this, the global methane concentration decreases by 25 ppbv.
This suggests that the coupling of both interactive hydrogen and methane may cause a decrease in the methane lifetime. Note, however, that these changes in methane lifetime are very small, and a more rigorous experiment should be conducted to confirm these results.

## 4.4   Pre-industrial and Present Day $H_2$

The pre-industrial simulation run from 1850 ($H_2$; $CH_4$-PI) has a much lower hydrogen burden of 129.4 Tg than present day
simulations due to having very low anthropogenic emissions. Despite this, the overall hydrogen lifetime is still in line with other present day simulations. This suggests that the increase in hydrogen emissions has a minimal impact on the loss terms, primarily due to soil uptake being independent of hydrogen in this model.

Figure 9 shows the hemispherically-averaged surface hydrogen and methane concentration from the a) PI ($H_2$; $CH_4$-PI) and b) present day (PD) simulation ($H_2$; $CH_4$-PD). The PI simulated surface methane concentration (Figure 9a) does not
have a strong gradient between hemispheres. There is a slightly greater concentration in the northern hemisphere than the south ($\approx 10$ ppbv). Using ice core data from Antarctica and Greenland, Etheridge et al. (1998) showed there was a 24 ppbv discrepancy between hemispheres in the 1850s; with southern hemisphere at 785 ppbv and northern hemisphere values at 809 ppbv. The $H_2$; $CH_4$-PI simulation has an annual average of 761 ppbv (SH) and 765 ppbv (NH). Both hemispheres have overlapping inter-annual variability (1 standard deviation), indicating there is no significant difference between hemispheres.
Hydrogen in the northern hemisphere has a strong seasonal cycle and ranges between 405 ppbv (March) to 360 ppbv (October) due to the strong deposition seasonality (as seen in Figure 2). In the southern hemisphere, the surface concentration remains consistent between $372 - 375$ ppbv. The southern hemisphere hydrogen concentrations can be compared directly with hydrogen concentrations reconstructed from firn air from Antarctica (Patterson et al., 2021). Around 1850s, Patterson et al. (2021) found that hydrogen levels were $330 \pm 15$ ppbv, which are slightly lower than the modelled concentration from the
$H_2$; $CH_4$-PI simulation. Firn air measurements for the northern hemisphere are only reconstructed back to 1950s, where the hydrogen concentration was $\approx 400 \pm 25$ ppbv (Patterson et al., 2023). The annually averaged $H_2$; $CH_4$-PI hydrogen concentration in the northern hemisphere is in agreement with the reconstructed firn data from 1950s. It is likely, however, that the





hydrogen concentration was lower in the 1850s, similar to that seen in the southern hemisphere. Thus, the simulated hydrogen concentration is likely overestimated.

There is a shift in the seasonal cycle of hydrogen in the PI simulation compared to the present day (Figure 9b; note the different axis for hydrogen concentration for PI and PD). In the northern hemisphere, hydrogen concentration peaks around April-May in PD run, while in the PI simulation, hydrogen peaks earlier in the year (March). This is a relative decrease in southern hemisphere emissions in the PI (likely due to reduced biomass burning) compared to the PD simulation. This is likely due to a decrease in soil uptake seasonality amplitude (PI: $4\,\mathrm{Tg}\;\mathrm{yr}^{-1}$ compared to PD: $9\,\mathrm{Tg}\;\mathrm{yr}^{-1}$), and a more consistent

production of hydrogen throughout the year (PI: $0.17\,\mathrm{mol}\;\mathrm{s}^{-1}$ range compared to $0.3\,\mathrm{mol}\;\mathrm{s}^{-1}$ range). This combination of a change in production and soil loss results in a shift in the northern hemisphere seasonality. Further to this, there is a shift in the hemispheric gradient (Figure 9c), with the hydrogen abundance in the northern hemisphere of the PI simulation greater than the southern hemisphere between December-July. The reduced interhemispheric gradient is due to a difference in the balance between northern and southern hydrogen emissions.

**4.5    Pulse Experiment**

In addition to the PI simulation, a present day hydrogen pulse experiment was conducted. Perturbation lifetime is a metric used to measure how much feedback a species would have on its own lifetime if its concentration increased. A feedback factor of 1 indicates that a species lifetime is independent of its own concentration. Perturbation lifetime is calculated by multiplying the atmospheric lifetime of hydrogen from the $H_2$; $CH_4$-PD by the feedback factor. In this experiment, the feedback factor is

calculated using the 6 years of decay from the $H_2$; $CH_4$-Pulse experiment:

$$\tau_P = \frac{\tau_L}{1-s} \tag{10}$$

where $\tau_P$ is the perturbation lifetime, $\tau_L$ is the steady state atmospheric lifetime from the $H_2$; $CH_4$-PD experiment, and $s$ is the sensitivity coefficient given by:

$$s = \frac{\delta(ln(\tau_L))}{\delta(ln(B))} \tag{11}$$

and $B$ is the burden of the whole atmosphere, and the $\delta(ln(\tau_L))$ used is the change in $H_2$ lifetime from the start of the $H_2$; $CH_4$-Pulse experiment to the end of the 6 years.

Hydrogen concentration at all altitudes and latitudes was set to $662.5\,\mathrm{ppbv}$ globally; a $25\%$ increase from the global-averaged surface concentration (see Methods section for more detail). Figure 10 shows the surface response of the system for 6 years after the pulse of hydrogen has been initiated. The dark blue line shows the response, while the green is the control. The

perturbation lifetime is calculated as 2.5 years, with a feedback factor of 1.04. This result is in agreement with Skeie et al. (2024), who also found that conducting hydrogen pulse emissions both across the globe and at specific locations did not have a significant impact on the hydrogen lifetime (and thus the global warming potential (GWP)). The calculated feedback factor



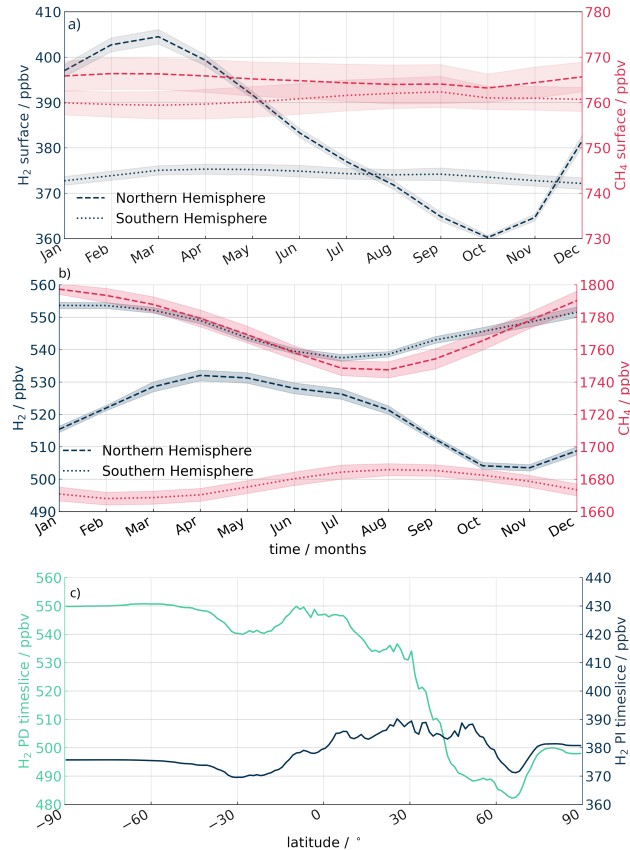

**Figure 9.** a) $H_2$; $CH_4$-PI and b) $H_2$; $CH_4$-PD simulations showing hemispherically-averaged hydrogen (blue) and methane (red) at the surface for 1850 (averaged over five years). Dashed lines show the northern hemisphere, while dotted lines show the southern. Shaded areas show one standard deviation calculated from the last five years of the simulation. c) Zonally averaged surface hydrogen concentration for (blue) PI and (green) PD simulations (averaged over give years).

is slightly above the average in Sand et al. (2023), who had a feedback factor of 1.0, but did not have interactive hydrogen and methane at the surface.

**5 Conclusions**

Hydrogen and methane are closely coupled species in the atmosphere, with both tracers competing for OH as their main atmospheric destruction pathway. We have implemented a hydrogen soil sink scheme into an ESM, along with adding the methane flux from Folberth et al. (2022) to create a fully coupled $H_2$-$CH_4$ ESM. Implementing $H_2$ flux caused the global average surface hydrogen to equilibrate to $540 - 550$ ppbv, which has been tuned to literature values of the tropospheric hydrogen burden




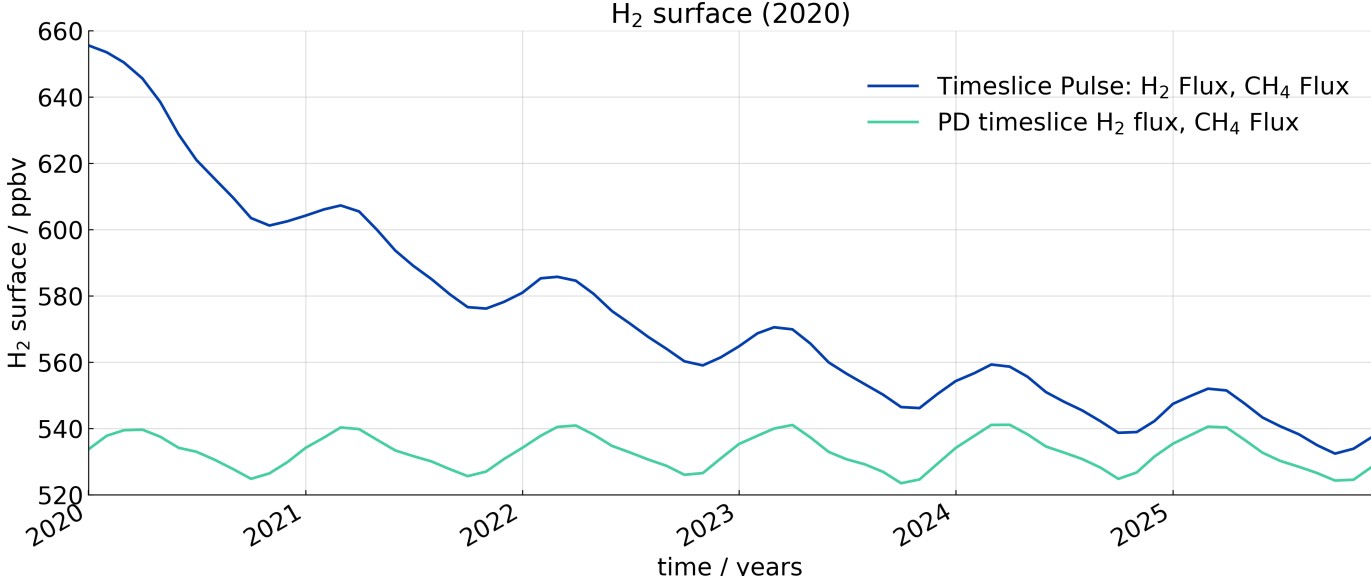

**Figure 10.** Global surface average of hydrogen concentration for (blue) pulse experiment and (green) 2020 timeslice and 6 years.

(Ehhalt and Rohrer, 2009). The model was able to capture the hydrogen seasonal cycle, which is dominant in the northern hemisphere and is primarily driven by the soil sink.

The inclusion of both $H_2$ flux and $CH_4$ flux had a minimal impact on the overall OH and ozone concentrations in the model, and there was little change in hydrogen concentration when $CH_4$ flux was added. Hydrogen and methane were compared against observations at a northern site (Mace Head) and a southern site (Cape Grim). Overall, both species were found to

capture the magnitude and the seasonal cycle of the observations when $H_2$ flux and $CH_4$ flux were on.

When implementing $H_2$ flux into the ESM with $CH_4$ flux, we found methane lifetime remained the same and the overall concentration of methane decreased. This occurred across all latitudes, with a larger decrease in the northern hemisphere. When $H_2$ flux and $CH_4$ flux are included, the ESM is able to simulation PI conditions which are within a similar order of magnitude as concentrations found in firn measurements. The seasonal cycle of hydrogen in the PI simulation shifted and peaked earlier

in the year, and the gradient between northern and southern concentrations was much smaller, with the hydrogen concentration in the north exceeding that of the south between December-July. Further to this, a hydrogen pulse experiment was conducted and a hydrogen feedback factor of 1.04 was found.

These initial experiments show the potential of a fully interactive hydrogen and methane model. Coupling $H_2$ flux and $CH_4$ flux into the ESM allowed the impact of hydrogen on methane to be analysed, and was found to cause a small global decrease

in methane. This unexpected behaviour highlights the importance of having a fully coupled hydrogen and methane model to further understand their interaction and, ultimately, provide a better estimate to their GWP in present day and future scenarios.



*Code and data availability.* THe NOAA methane and hydrogen data is available from https://www.gml.noaa.gov/dv/iadv/ (last accessed: 5th June, 2025). Data from all simulations is found at Brown (2025).

All simulations used in this work were performed using version 12.0 of the Met Office Unified Model coupled to the United Kingdom Chemistry and Aerosol model (UM–UKCA).The UM code branch used in the publication have not all been submitted for review and inclusion in the UM trunk or released for general use. However, the UM and JULES code branches were made available to reviewers of this paper. Due to intellectual property copyright restrictions, we cannot provide the source code for UM–UKCA. The UM–UKCA model is available for use through a licensing agreement. A number of research organisations and national meteorological services use UM–UKCA in collaboration with the Met Office to undertake basic atmospheric process research, produce forecasts, develop the model code, and build and

evaluate Earth system models. Please visit https://www.metoffice.gov.uk/research/approach/modelling-systems/unified-model (last accessed: 5th June 2025) for further information on how to apply for a licence.

## Appendix A

### A1    Hydrogen Deposition Scheme

The equation for hydrogen deposition, $v_d(\mathrm{H_2})$ (also known as $\frac{1}{\mathrm{rc}}$), is given by,

$$v_d(\mathrm{H_2}) = \frac{1}{\frac{\delta}{D_s} + \frac{\delta_{\mathrm{snow}}}{D_{\mathrm{snow}}} + \frac{1}{\sqrt{D_s k_s \Theta_a}}} \tag{A1}$$

    The deposition scheme can be divided up into three terms; the first describes soil diffusivity where $\delta$ is the soil depth, and $D_s$ is the gas diffusivity of hydrogen into the soil. The second is similar, but describes the diffusion through snow; $\delta_{\mathrm{snow}}$ is the snow depth and $D_{\mathrm{snow}}$ is the gas diffusivity of hydrogen through snow. The third term, $k_s \Theta_a$, describes the microbial uptake and can be split into multiple equations. The gas diffusivity of hydrogen is given as:

$$D_s = \frac{\Theta_a^{3.1} \cdot D_a}{\Theta_p^2} \tag{A2}$$

$\Theta_a$ is the volumetric air fraction (m$^3$ air / m$^3$ total pore space), $\Theta_p$ is the soil porosity (m$^3$ total pore space / m$^3$ total volume). $D_a$ is given by:

$$D_a = 0.611 \cdot \frac{1013.25}{p} \cdot \left( \frac{T + 273.15}{273.15} \right)^{1.75} \tag{A3}$$

$p$ is surface pressure (hPa) and $T$ is soil temperature (C$^\circ$). $k_s \Theta_a$ can be broken down into three functions:

$k_s \Theta_a = A \cdot f(\Theta_a) \cdot g(T_s)$                                           (A4)

    $A$ is a scaling factor and is used as a proxy for microbial activity:





$$A = \alpha \frac{\text{soil}_C}{\beta + \text{soil}_C} \qquad (A5)$$

where $\text{soil}_C$ is the soil carbon content and $\beta = 7\,\text{kgC m}^{-3}$ as per Paulot et al. (2021). $\alpha$ is a unitless parameter, which was set to 30 by scaling the tropospheric hydrogen global burden to approximately $155\,\text{Tg}$ and, thus, inline with results from Ehhalt and Rohrer (2009). The $g(T_s)$ term from Equation A4 is given by:

$$g(T_s) = \frac{1}{1 + \exp(-\frac{T-3.8}{6.7})} + \frac{1}{1 + \exp(\frac{T-62.2}{7.1})} - 1 \qquad (A6)$$

while the $f(\Theta_a)$ function is dependent on the soil type. For sand, the equation is given by:

$$f_s(\Theta_a) = 0.00936 \cdot \frac{(\frac{\Theta_w}{\Theta_p} - 0.02640) \cdot (1 - \frac{\Theta_w}{\Theta_p})}{(\frac{\Theta_w}{\Theta_p})^2 - 0.1715 \cdot (\frac{\Theta_w}{\Theta_p}) + 0.03144} \qquad (A7)$$

where $\Theta_w$ and $\Theta_p$ are the volumetric soil moisture content (SMC) and the total porosity respectively, with limits;

$$\Theta_p \geq 1 \qquad (A8)$$

$$\Theta_w \geq 0.0264 \qquad (A9)$$

For loam, $\Theta_a$ is:

$$f_l(\Theta_a) = 0.01997 \cdot \frac{(\frac{\Theta_w}{\Theta_p} - 0.05369) \cdot (0.8508 - \frac{\Theta_w}{\Theta_p})}{(\frac{\Theta_w}{\Theta_p})^2 - 0.7541 \cdot (\frac{\Theta_w}{\Theta_p}) + 0.2806} \qquad (A10)$$

The limits for $f_l(\Theta_a)$ are:

$$\Theta_p \geq 0.8511 \qquad (A11)$$

$$\Theta_w \geq 0.0537 \qquad (A12)$$



**Table A1.** NOAA observation site for hydrogen divided into north and south. Numbers correspond to the values even in the Taylor diagram (Figure 5). Values with * do not appear on the Taylor diagram due to either having a negative correlation, or a normalised standard deviation greater than 2

| Site number (N) | Name (N) | Site Number (S) | Name (S) |
|---|---|---|---|
| 1 | h2 ask surface-flask 1 ccgg event | 1 | h2 crz surface-flask 1 ccgg event |
| 2 | h2 bal surface-flask 1 ccgg event | 2 | h2 bkt surface-flask 1 ccgg event* |
| 3 | h2 llb surface-flask 1 ccgg event | 3 | h2 cpt surface-flask 1 ccgg event |
| 4 | h2 dsi surface-flask 1 ccgg event | 4 | h2 asc surface-flask 1 ccgg event |
| 5 | h2 brw surface-flask 1 ccgg event | 5 | h2 bhd surface-flask 1 ccgg event |
| 6 | h2 pal surface-flask 1 ccgg event | 6 | h2 sey surface-flask 1 ccgg event |
| 7 | h2 nwr surface-flask 1 ccgg event | 7 | h2 psa surface-flask 1 ccgg event |
| 8 | h2 wis surface-flask 1 ccgg event | 8 | h2 drp shipboard-flask 1 ccgg event |
| 9 | h2 sdz surface-flask 1 ccgg event* | 9 | h2 mkn surface-flask 1 ccgg event* |
| 10 | h2 bsc surface-flask 1 ccgg event | 10 | h2 spo surface-flask 1 ccgg event |
| 11 | h2 sgp surface-flask 1 ccgg event | 11 | h2 hba surface-flask 1 ccgg event |
| 12 | h2 azr surface-flask 1 ccgg event | 12 | h2 nat surface-flask 1 ccgg event |
| 13 | h2 hsu surface-flask 1 ccgg event | 13 | h2 nmb surface-flask 1 ccgg event* |
| 14 | h2 uum surface-flask 1 ccgg event | 14 | h2 smo surface-flask 1 ccgg event |
| 15 | h2 tik surface-flask 1 ccgg event | 15 | h2 cgo surface-flask 1 ccgg event |
| 16 | h2 thd surface-flask 1 ccgg event | 16 | h2 ush surface-flask 1 ccgg event |
| 17 | h2 cib surface-flask 1 ccgg event | 17 | h2 syo surface-flask 1 ccgg event |
| 18 | h2 uta surface-flask 1 ccgg event | 18 | h2 eic surface-flask 1 ccgg event |
| 19 | h2 chr surface-flask 1 ccgg event | - | |
| 20 | h2 mlo surface-flask 1 ccgg event* | - | |
| 21 | h2 key surface-flask 1 ccgg event | - | |
| 22 | h2 izo surface-flask 1 ccgg event | - | |
| 23 | h2 wpc shipboard-flask 1 ccgg event | - | |
| 24 | h2 poc shipboard-flask 1 ccgg event | - | |
| 25 | h2 oxk surface-flask 1 ccgg event* | - | |




**Table A1.** Cont.

| Site number (N) | Name (N) | Site Number (S) | Name (S) |
|---|---|---|---|
| 25 | h2 oxk surface-flask 1 ccgg event* | - | |
| 26 | h2 alt surface-flask 1 ccgg event* | - | |
| 27 | h2 bmw surface-flask 1 ccgg event | - | |
| 28 | h2 rpb surface-flask 1 ccgg event | - | |
| 29 | h2 hun surface-flask 1 ccgg event* | - | |
| 30 | h2 tap surface-flask 1 ccgg event | - | |
| 31 | h2 shm surface-flask 1 ccgg event | - | |
| 32 | h2 mex surface-flask 1 ccgg event | - | |
| 33 | h2 sum surface-flask 1 ccgg event | - | |
| 34 | h2 ice surface-flask 1 ccgg event | - | |
| 35 | h2 hpb surface-flask 1 ccgg event | - | |
| 36 | h2 lmp surface-flask 1 ccgg event | - | |
| 37 | h2 pta surface-flask 1 ccgg event | - | |
| 38 | h2 kum surface-flask 1 ccgg event | - | |
| 39 | h2 cba surface-flask 1 ccgg event | - | |
| 40 | h2 mid surface-flask 1 ccgg event | - | |
| 41 | h2 mhd surface-flask 1 ccgg event | - | |
| 42 | h2 wlg surface-flask 1 ccgg event | - | |
| 43 | h2 zep surface-flask 1 ccgg event | - | |
| 44 | h2 lln surface-flask 1 ccgg event | - | |



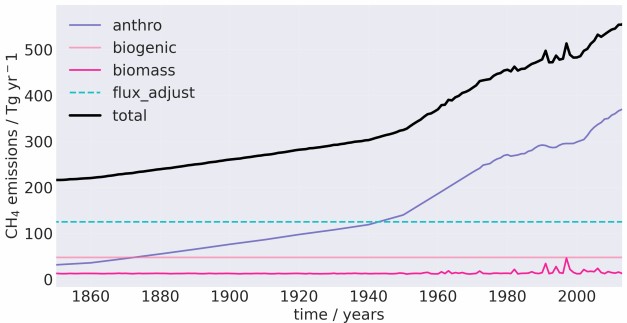

**Figure A1.** Yearly, globally averaged methane emissions from $1850 - 2014$ (excluding wetlands). The flux adjustment is that used in the nudged simulations and is a total $124$ CH$_4$ Tg yr$^{-1}$.



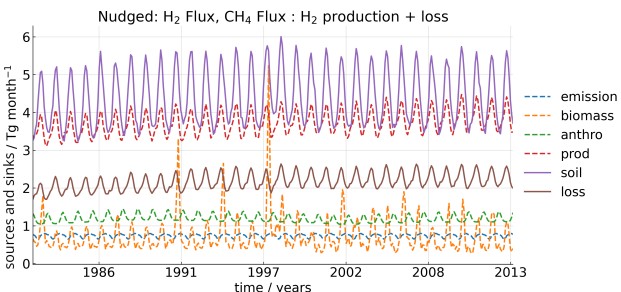

**Figure A2.** Globally averaged hydrogen sources and sinks from $1982 - 2013$).



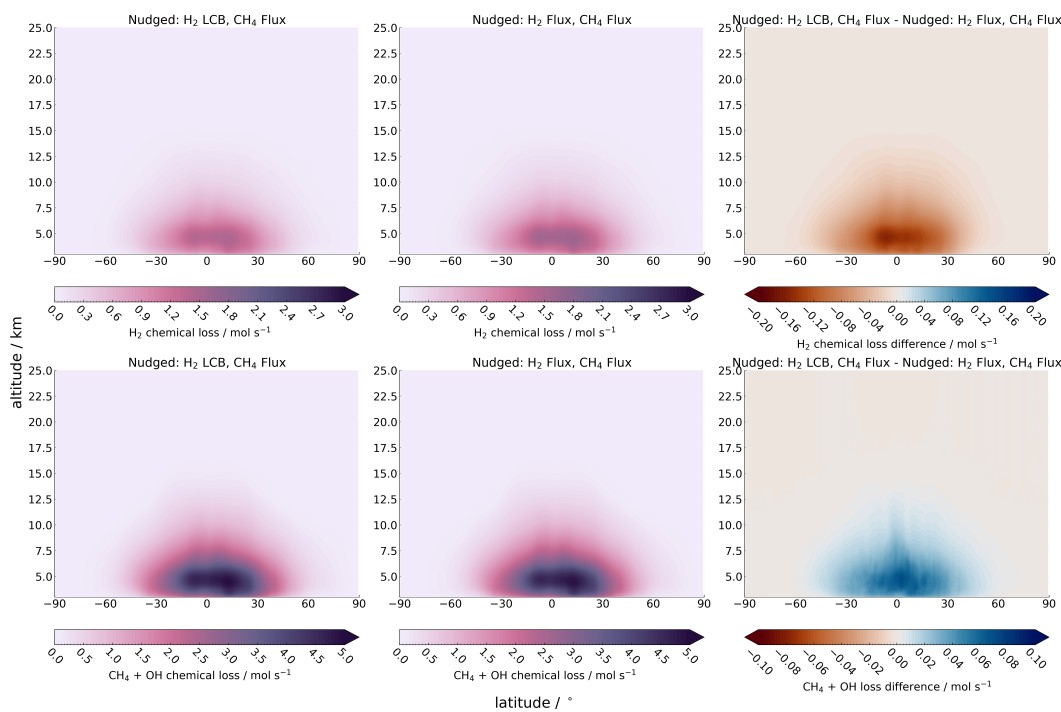

**Figure A3.** Zonally averaged (top) $H_2$ chemical loss and (bottom) $CH_4$ loss via OH for (left) $CH_4$-flux simulation, (middle) $H_2$; $CH_4$-flux simulation, and (right) difference between the two ($CH_4$-flux $-$ $H_2$; $CH_4$-flux) averaged between $2008-2013$. Red (blue) shows an increase (decrease) from the $CH_4$-flux. Note the non-linearity of scales.





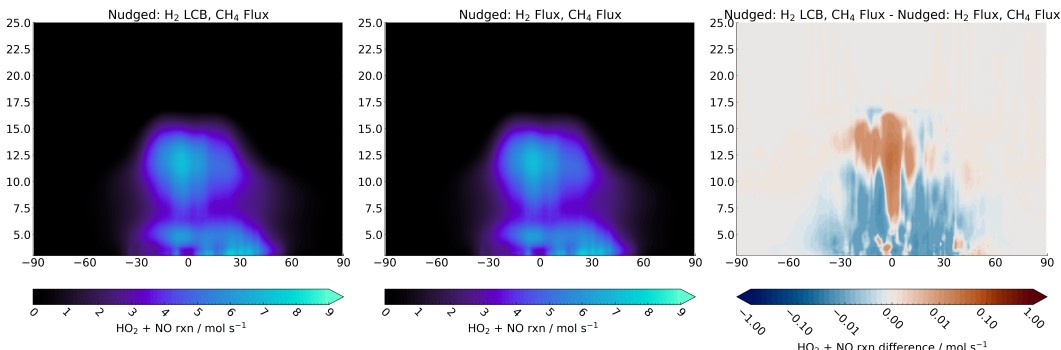

**Figure A4.** Zonally averaged reaction rate of k[HO$_2$][NO] for (left) CH$_4$-flux simulation, (middle) H$_2$; CH$_4$-flux simulation, and (right) difference between the two (CH$_4$-flux − H$_2$; CH$_4$-flux) averaged between 2008 − 2013. Red (blue) shows a increase (decrease) from the CH$_4$-flux. Note the non-linearity of scales.



*Author contributions.*  MAJB ran the simulations, analysed the data and wrote the manuscript. ATA and NJW helped with the analysis, ideas, and proofread the manuscript. PTG helped with ideas and proofread the manuscript. NLA set up and provided the model runs for all simulations and proofread the manuscript. STR produced the hydrogen emissions needed for $H_2$ flux simulations and contirubted to the

methods section. GAF and FMO provivded the $CH_4$ flux emissions files setup, helped with analysis and proofread the manuscript.

*Competing interests.*  There are no competing interests present in this work.

*Acknowledgements.*  MB, NJW, NLA, and ATA were supported by the National Environment Research Council through grand NE/X010236/1. F.M. O'Connor and G.A. Folberth were supported by the Met Office Hadley Centre Climate Programme funded by DSIT and through the EU Horizon project ESM2025 (Grant 101003536). This work used JASMIN, the UK collaborative data analysis facility. This work used the

ARCHER2 UK National Supercomputing Service (https://www.archer2.ac.uk) (Beckett et al., 2024). We acknowledge use of the Monsoon2 system, a collaborative facility supplied under the Joint Weather and Climate Research Programme, a strategic partnership between the Met Office and the Natural Environment Research Council.



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
