# Peer review of "Development of Fully Interactive Hydrogen with Methane in UKESM1.0"

_EGUsphere, 2025_

## Author Comment (AC1)

*We thank the editor and both reviewers for the time and efforts they have given. We have incorporated all of the comments from both reviewers, and we believe has significantly improved the manuscript.*

*Line numbers refer to the lines of the original manuscript. Our replies to comments are given in green.*

*We have made several large changes to the manuscript, which we have outlined below.*

*We have run two additional timeslice simulations to analyse the methane response to hydrogen when hydrogen emissions are perturbed and both fluxes are turned on. We can confirm that the methane response to hydrogen in a flux-flux interaction is in agreement with the rest of the current literature. This is discussed in Section 4.2. After further analysis, we have modified the section describing the potential chemical reactions in Section 4.2, which we believe now gives a more coherent and plausible explanation.*

*We decided to remove Section 4.5 (the Pulse Experiment) as we realised the pulse simulation would not allow for an analogous calculation of perturbation lifetimes comparable with the literature, and it may otherwise cause confusion and does not directly relate to the H2-CH4 interaction. We had previously answered all reviewers' comments on Section 4.5, however we have since removed these from our responses.*

—----------------------------------------------

**Reviewer 1 Comments**

Summary

The manuscript describes a new configuration for UKESM1 to simulate flux boundary conditions for both H2 and CH4 at the same time, with the goal to more accurately model their interactions. A number of sensitivity simulations were conducted to isolate effects of switching from fixed lower boundary conditions (LBCs) to fluxes for either or both of the species. While the results of simulations with flux boundary conditions for the recent past broadly agree with measurements, the authors find a reduction in CH4 abundance associated with an increase in surface H2 mixing ratio, which is in contrast to all recent publications on effects of additional H2 in the atmosphere (e.g., Paulot et al., 2021, https://doi.org/10.1016/j.ijhydene.2021.01.088; Warwick et al., 2023, https://doi.org/10.5194/acp-23-13451-2023; Sand et al., 2023, https://doi.org/10.1038/s43247-023-00857-8).

To showcase applications of the model, a pre-industrial and a hydrogen pulse experiment were run and analyzed. A comparison of CH4 and H2 levels from the former to observation-derived data served as further evidence of the model's applicability, while

the latter was used to determine a feedback factor for an atmospheric hydrogen increase.

General comments

The manuscript is generally well structured and written, the content clearly falls within the scope of GMD, and is certainly of sufficient scientific novelty. I strongly encourage its publication, but I think it requires some additional explanations.

We thank the referee for their detailed and helpful comments. We are pleased they find the work is of "sufficient scientific novelty" and that they "strongly encourage publication". We have responded to their specific and general points below.

1. After carefully re-reading it several times, I still struggle with the question whether or why a change from a fixed LBC for H2 to emission and deposition fluxes should change the response of the CH4 abundance to additional H2---or why a model with CH4 flux boundary conditions instead of a fixed LBC should react differently to additional H2. This should be addressed (more clearly) in my opinion before a study including applications of the new model configuration is published in ACP.

   We are slightly unclear why ACP has been referenced here. We take it in the more general sense that the reviewer finds that there is a need for the full scientific reasons behind the difference in response to be outlined here, in this GMD(D) paper, before follow up studies are published in other journals (such as ACP). However, we note that the scope of GMD is such that this submission (a technical development ) does not require this. Moreover, based on feedback from the referees we conducted further experiments that we feel address the major uncertainties that the referees brought up re the scientific interpretation of our results. We have i) cut down the discussion on the LBC vs Flux experiments (adding a cautionary note for others) ii) added in a new section on the Flux perturbation runs and in doing so feel we have addressed all scientific queries the referees had.

2. Especially, the changes induced by switching from fixed LBCs to fluxes should be disentangled from concurrent changes in the CH4 and H2 abundances. If resources permit, this could be achieved with the help of additional experiments with fewer changes, e.g., replacing fluxes by fixed LBCs derived from the corresponding flux-driven simulations, or repeating the PD timeslice simulation with different fixed LBCs, or tuning the H2 soil sink to match the H2 burden from the fixed LBC simulations. All budget terms could then be compared between the simulations, to illustrate and explain the interplay of the different processes.

   Firstly, we are unable to do a like for like budget analysis on the model simulations as we did not diagnose fluxes through the LBC which would be required for budget closure. Several of the experiments are pulses where the budget changes dramatically over time and so would not be directly comparable

with the literature. Hence, we feel that sticking to the budgets as presented in Table 2 is sufficient for the purposes of this study.

We have been able to run a further set of simulations where we have perturbed the emissions of H2 by x1/4 (decrease in H2) and x4 (increase in H2) in a timeslice configuration. We've run three simulations, including a control run with unchanged H2 emissions. Both the H2 and CH4 are interactive in these runs. We found that when increasing the amount of H2 in the model, the CH4 also increases and is in agreement with Warwick et al. (2022). We ran the model for an additional 10 years, which only corresponds to ~1 CH4 lifetime cycle. The x4 run shows the exaggerated response of CH4 to H2 to dissolve any ambiguity.

In addition, we have removed the section on OH recycling to create more OH as we found a more plausible explanation from the additional runs (lines 230-238 and 246-264).

We have added a small section to highlight the reason for the change in behaviour when comparing runs with LBC and verifying the H2-CH4 interaction when the H2 budget changes.

Altered line 35:

"We then run the fully interactive ESM under pre-industrial conditions and with sustained increased and reduced hydrogen emissions to analyse the methane response to changes in hydrogen under present day scenarios."

Methods (after line 54):

"While the LBC and interactive simulations offer a step by step response with and without interactive H2 and CH4 in a nudged scenario, these simulations are not directly comparable due to the differences in treatment of CH4 and H2 at the surface. In order to make an equal comparison, two additional timeslice simulations were run over the year 2020. The TS-H2CH4-PD run is used as a control, while in the other two simulations all H2 emissions are continuously multiplied by 1/4 (TS-H2CH4-0.25) and 4 (TS-H2CH4-4).These simulations are run for ten years to assess the response of CH4 to different levels of H2. Similarly to the pulse experiment, the same starting conditions as the TS-H2CH4-PD simulation were used. Table 1 summarises these simulations."

We have adapted line 230:

"One explanation for this is due to the change in surface distribution for H2 mixing ratio which impacts the OH reactivity. This is a result from the technical differences in simulations (e.g. interactive flux vs fixed LBC). Figure 8 shows the one over the rate coefficient of (top) H2 chemical loss and (bottom) CH4 loss via OH ($1/(k[OH])$), which shows the OH reactivity of these reactions, and simultaneously the respective chemical lifetimes of H2 and CH4. Blue (red)

indicates a decrease (increase) in lifetimes when H2 flux is included, relative to the simulation with fixed H2 LBC.

The surface hydrogen mixing ratio in the southern hemisphere increases up to 580 ppbv (as seen in Figure 5b), while decreasing in the northern hemisphere down to 480 ppbv. The shift in hydrogen distribution causes a change in OH reactivity. In the southern hemisphere, the CH4 lifetime via OH increases by up to 0.4%, which can be seen at the surface between 30-60 S (red) in bottom right panel Figure 8. In the northern hemisphere, the CH4 lifetime via OH decreases by ~0.3% (blue in top right panel of Figure 8). This is a result of the reduction in H2 mixing ratio at the surface in the northern hemisphere, which is directly due to the stronger soil uptake over northern American and Siberia."

The different surface set ups (fixed LBC versus interactive flux) are not a fair direct comparison of the response of CH4 to H2. To examine the impact of a change in H2 on CH4 without the added complication of the fixed LBCs, three additional timeslice simulations were analysed; a control (TS-H2CH4-PD), one with reduced (25% reduction; TS-H2CH4-0.25) H2 emissions, and another with increased (400% increase; TS-H2CH4-4) H2 emissions.

Figure 9 summarises the globally averaged surface mixing ratios of (a) H2 and (b) CH4 for 10 years. Figure 9c shows the monthly, globally average surface difference between the control and the altered emissions simulations (control - alter emissions). The black line shows there is no change from the control. The TS-H2CH4-4 simulation shows that CH4 increases as H2 increases. Figure 9c shows a non-linear relationship with the x4 H2 emission simulation. This is likely due to H2 reaching steady state much quicker than CH4 as its lifetime is shorter (2 years compared to 10 years), while CH4 continues to increase in response to H2 (Figure 9c)."

Also in line 280:

"This difference in global surface methane mixing ratio is due to the spatial distribution of H2 from the different methodologies (e.g. interactive flux and fixed LBC) and is therefore difficult to directly compare against.

Table 3 also shows the CH4 lifetime via OH loss for the present day timeslices with adjusted H2 emissions.

The TS-H2CH4-0.25 simulation which has lower H2 emissions is 0.12 years lower than the present day control, while the TS-H2CH4-4 with increased H2 emissions has an increased CH4 lifetime of 0.43 years.

Given that these differences are larger than in the nudged simulations and are not obscured by different methodologies (all present day timeslices have both H2 and CH4 interactive flux), the response of CH4 to an increase in H2 is in agreement with Warwick et al. (2023) and our general scientific understanding.

Moreover, the changes seen in CH4 in the nudged simulations are due to different methodologies of surface interaction and not the response of an increase in H2." [Note that Table 3 has been updated, along with its caption]

And line 347:

"When implementing H2 flux into the ESM with CH4 flux, we found methane lifetime remained the same and the overall global surface mixing ratio of methane decreased. This decrease is due to the change in the spatial distribution of hydrogen at the surface when switching from fixed H2 LBC to H2 flux, the former of which has no spatial variability. With the replacement of H2 LBC to H2 flux, the overall global surface hydrogen mixing ratio increased. However, the hydrogen mixing ratio decreased in the northern hemisphere as a result of the soil sink, which led to a decrease in the CH4 lifetime via OH."

And finally in line 353:

"A further set of experiments perturbing H2 emissions in a present day timeslice show a similar response as Warwick et al. (2023), where an increase in H2 results in an increase of CH4. The small decrease in CH4 in the nudged simulations when replacing a fixed H2 LBC with a H2 flux is, therefore, due to comparing different methodologies of surface interactions (interactive flux versus fixed boundary layer)."

3. It may also be possible to derive a more detailed understanding from the available data (e.g., analysis of OH and CH4 in the hydrogen pulse experiment, and comparison to the PD timeslice), but the currently presented analysis in Sect. 4.2 does not convince me. Rather, I agree with the authors' own conclusion at the end of Sect. 4.3 that "a more rigorous experiment should be conducted to confirm these results", as they have the potential to drastically change the view on all the recent studies on climate effects of hydrogen emissions. The main question seems to be whether the net effect of additional H2 on OH abundance, (-)dOH/dH2 in the notation of Warwick et al., 2023, https://doi.org/10.5194/acp-23-13451-2023, is an increase or a decrease, or whether there are different regimes, and which of them is prevalent under which conditions.

Please see the response above.

**Specific comments**

Depending on final layout, consider reducing the number of figures to ease readability by bringing the figures closer to their discussion.

4. l. 26: Please specify what you mean with "the methane feedback factor from the impact of hydrogen".

We have changed this to: "will give a better understanding of the role hydrogen plays in the methane feedback factor."

5. l. 40: Please explain how the fixed LBCs are implemented. (Overwrite or "nudge"? Only lowest layer?)

We have added the following on line 40 to clarify: "The LBC resets the value of the relevant species at the lowest vertical level after each timestep."

6. l. 43f: Please explain and/or add a reference for the "very limited interannual variability" of hydrogen deposition. Since soil moisture plays a crucial role, I would expect some degree of variation, at least in the midlatitudes.

We have rephrased this sentence:

"The Sanderson scheme was only run for one year as there is only very limited interannual variability in the forcing soil temperature and soil moisture data from the land surface model, JULES (Pinnington et al 2018)."

7. l. 48: for how many years are the time slices run after the spin-up?

Clarified by the following: "Data are analysed over a 5-year period after the initial 25 year spin up, to allow for methane to reach a suitable steady state after ~2.5 methane lifetimes."

8. l. 61: Since you (probably) used the same datasets for anthropogenic and biomass burning as Paulot et al., 2021, https://doi.org/10.1016/j.ijhydene.2021.01.088 did, why did you derive emission ratios from their Table 1 values instead of simply applying the ratios (and emission factors, in case of BB) they supply in their supplement? Please make sure that you arrived at the same values. Furthermore, the oceanic and terrestrial emissions should be excluded from the parenthesis in l. 61, as they are not associated with CO emissions, if I am not mistaken.

Modelled CO emissions will be complicated by the fact that most models add in to the direct CO emission a component that represents missing reactivity from unrepresented non-methane volatile organic compounds (e.g., Archibald et al., 2020). Rather than apply the emissions from Paulot et al. (2021) it is necessary to construct the emission ratios and apply these to build a model specific emission inventory.

Indeed, the scaled H2 emissions do match up with those from Paulot et al. We have adapted H2 emissions in Figure A2 to show Tg yr-1 and therefore more easily comparable Paulot et al. and with other studies. We have also rephrased the emissions:

"The resultant hydrogen emission for anthropogenic and biomass burning sources follow the spatial pattern of the equivalent CO source, but with values rescaled to give the global emission total appropriate for hydrogen. Scalings for

oceanic and terrestrial H2 emissions were set as 6 and 3 Tg respectively with the spatial distribution following that of CO as in Paulot et al. (2021). The H2 emissions are in agreement with Paulot et al (2021) as shown in Figure A2."

9. l. 88: What is "[t]his larger flux adjustment" here? From the next line, I would assume it refers to the difference between 135 Tg yr-1 and 190 Tg yr-1, but at the end of the paragraph, the text says that "the" flux adjustment from the PD time slice was increased by(?) 20% (although it suggested previously that you calculated a new flux adjustment). Furthermore, how do the 20% relate to the ~40% difference in the wetland emissions? Are there separate adjustments for the wetland and other emissions? I did not consult Folberth et al., 2022, https://doi.org/10.1029/2021MS002982, for answers to these questions, but I think the explanation here should be revised to not require reading that publication first. In my opinion, the whole description of the flux adjustment for the nudged simulations should be revised. Furthermore, please make sure the 124 Tg yr-1 in the caption of Fig. A1 are also related to this description.

We have since rewritten this section, see R2 comment #13. We have also added in a reference to Figure A1.

10. l. 97: Which other processes are considered in the top layer?

We have rephrased to clarify: "The first layer represents the diffusion of hydrogen through the top layer of soil..."

11. l. 104f: What is the motivation for this discussion (and Figure 1)? My feeling is that neither is necessary.

Previous studies have excluded the other soil resistances as they found them negligible. Here we describe that aerodynamic and laminar resistances are included and describe how they impact the total soil deposition. We have added the following to clarify its importance in line 103: "Different hydrogen deposition schemes vary in the inclusion of other deposition resistances (e.g. Bertagni et al. (2023)). In this model, aerodynamic (ra) and laminar (rb) resistances are incorporated."

12. l. 123: If you describe the soil porosity as the likely inaccuracy causing too high SMC/porosity ratios, why do you adjust SMC and not porosity?

SMC has been derived from porosity - if we changed only the porosity, the model would break and the SMC values would no longer be valid. Our aim isn't to change the soil porosity as this would impact many other variables in the land-surface model, but rather to make a correction to the deposition while having minimal impact on the rest of the model.

13. Fig. 3: Depending on the values, showing the standard deviations from the simulation across the aggregation intervals may add further value to the comparison (should then be discussed in the text as well, of course).

As some observations are only for one year, we cannot add on standard deviations for all the sites. As for the other simulations with multiple years, we did originally have the standard deviations on, but found they were very small as there was very little interannual variability.

14. Fig. 3 caption and description in text: Please add information on data selection/aggregation for the figure (monthly means for the years of the measurements?).

We have added the following in the caption: "Monthly mean observed hydrogen deposition velocities averaged across years where possible (points) compared with…"

15. Fig. 3 discussion:

    a. From the Figure, I would conclude that the Sanderson et al. parametrization results are no worse, if not better, than the new approach when it comes to the comparison with observations. Please comment on (or emphasize more) why you still chose to replace the Sanderson et al. scheme entirely, instead of maybe adapting it where it has its shortcomings.

The Sanderson scheme does a good job of producing deposition velocities but it has limitations. Namely that it is dependent on land use and not functions of soil properties. This means it is constant with time and so cannot be used for time dependent studies (future or pre-historical predictions).

Furthermore it has no uptake over dry regions such as deserts (regardless of soil moisture). These are outlined in lines 125-129.

    b. Please comment on the differences between your comparison and the one by Paulot et al., 2021, https://doi.org/10.1016/j.ijhydene.2021.01.088, e.g., availability of summer data for Gif-sur-Yvette; differences in Sanderson parametrization results, especially for Mace Head; why your comparison does not

The differences between schemes used in this model and GDFL from Paulot et al. 2021 are due to different input variables from their respective land-surface models. E.g. Soil porosity is different, so soil moisture is different (this impacts the Sanderson scheme), and the land use will also be distributed differently. The two-layer scheme in each model is further tuned through the A parameter. We believe the hydrogen deposition of GDFL has already been accounted for by including the CMIP6 range (shaded area) of which GDFL is one of the GCMs included.

Belviso et al (2013) did not provide data during the summer months for the Gif-sur- Yvetter site, so we were unable to compare to these.

    c. l. 150: Please rephrase or expt include the other three stations that Paulot et al. considered; etc.lain what you mean with "averaged over one year" for the Sanderson simulation.

We did not use as many sites as in Paulot et al. (Tsukuba, San Jacinto Mountain Reserve, and Heidelberg) as this data is not provided in the original papers. We have removed the "averaged over one year" phrase.

16. ll. 170f: Please add justification for this hypothesis, e.g., actually evaluate the soil moisture that you simulate for Tasmania by comparison with satellite observations.

  We have added a reference to support this

17. Fig. 4 discussion:
    a. While I agree that the (absent) trend in [H2] at CGO is captured well by the model from about 2003 onwards, I miss a comment on the disagreement during the period 1994--2003, which may also be reflected -- although to a lesser degree -- in the MHD comparison, where agreement is reached around 2002.

We have added the following to line 168: "The simulated hydrogen captures the trend post 2003 at Cape Grim, but does not do as well prior to this."

    b. Why does the near-perfect agreement between simulated and observed [CH4] at CGO break after 1992? Can this be due to the way the flux correction is determined?

This occurs when the growth of methane slows down/pauses for several years and is difficult for GCMs to capture this change in trend. The flux correction is static across years (this has now been clarified in Section 2.3 - see reply to Reviewer 2, comment #13).

For the nudged simulation, the flux adjustment used in the PD timeslice simulation was scaled to enable the model to capture the growth in surface global mean methane during the 1980s. This helps explain why the model performs well at CGO up until 1992. However, the inclusion of that time-invariant flux adjustment contributes to an overestimate at CGO in later years. More generally, models struggle to capture methane trends over the past few decades using methane emission inventories. There could be a number of reasons for this, which are not in the scope of this study.

    c. Similarly, how do you explain the large reduction in deviation between the simulated and observed [CH4] at MHD between 1992 and ~2002?

As we say above, flux adjustment was scaled to enable the model to capture the growth in surface global mean methane during the 1980s. It was not tuned to specific locations. In the case of Mace Head, the inclusion of this flux adjustment has contributed to an overestimate in methane prior to 2000. Determining specific causes for differences between simulated and observed methane mixing ratios does not form part of the scope of this study.

18. Comparison to NOAA [H2]: Unless I miss an important detail, please exclude the year 2014 from the observations as well, as you only simulated until 2013.

Agreed. This was an error in writing and has now been corrected

19. l. 203: Please elaborate on the effect of orography and how the Hayman et al., 2014, https://doi.org/10.5194/acp-14-13257-2014, publication helps explain this [H2] overestimation.

We have added the following to clarify in line 203: "...and orography in the model which is known to lead to an overestimation in emissions (Hayman et al. 2014)."

20. ll. 205f: An underprediction of <0.4% can be explained by anything in my opinion. Thus, I recommend either to remove the association with deposition here (and also the last sentence of Sect. 3), or to report maybe the median deviation rather than the mean, which may be quite strongly biased by the large deviations over China.

Done. We have removed this sentence.

21. Sect. 4.1: Please add a comment on the very high simulated [H2] over central/tropical Africa as well, where the sink is actually quite strong as well, according to your Fig. 2.

H2 values are high over these regions due to biomass burning. We have added the following in line 201: "Hydrogen concentration is high (>580 ppbv) in Africa between 5N-25S due to biomass burning and natural emissions, despite the larger deposition velocity over that region."

22. Fig. 6 discussion/image: changes mixed up between c_OH (opposite changes in troposphere and stratosphere in the figure) and [O3] (rather homogeneous difference in the figure)?

Thank you for pointing this out - we've revised the text to change this:

"There is up to a 40 ppbv change in ozone between the control and H2;CH4 flux simulation, which corresponds to a 4% decrease. Similarly, there is less than 5e4 molecules cm-3 increase in OH density in the troposphere (3% increase), and a decrease of 4e4 molecules cm-3 in the stratosphere (5% decrease)."

23. l. 226f: While I agree that the trend until 1992 is captured well, the simulated [CH4] levels off much more quickly and stably than in the observations, and the increasing trend after 2007 is much steeper. Please explain these differences.

For a full analysis of the methane trend, we refer the reviewer to Folberth et al. (2022) who did this study. We do not go into detail about the methane trend in this work as the purpose of this manuscript is to focus on the H2-CH4 interaction.

24. l. 228: The difference in [CH4] between the simulations with fixed H2 LBC and H2 fluxes seems to grow over time. Considering the long methane lifetime, such behavior is of course expected, but I find it hard to judge from the figure whether it is not still increasing at the end of the three simulated decades. Please discuss this in the text.

We plotted the difference in methane between the two runs which showed the CH4 difference levelling out around 1998. We have added in the following in line 228; "...the global, surface average methane is approximately 20 ppbv lower from 1998 onwards."

25. l. 247: How do you infer the causality here?

We have now removed this section (see R1 comment #2 for more details)

26. Fig. 7: Please explain (or fix) the data gap in the surface [CH4] from the H2;CH4-flux simulation in 1986.

Fixed.

27. Fig. 8 and its discussion: Please explain the relevance of differences of the order of 1e-4, if you consider even the 2% change in CH4 lifetime "minimal" (ll. 272f).

We have changed the scale to a non-logarithmic scale to show the differences and avoid any potential confusion. The changes are now up to 1%, which are still very small. We also add in line 247: "While these values are small (up to 1% difference), the spatial and vertical locations of these differences match between the CH4 and H2 chemical loss rates."

28. Fig. 8: I think it is not good style to leave out data from a figure. Maybe the "noise" (I assume you mean large relative differences and their fluctuation) could also be reduced by plotting the inverses of the lifetimes (i.e., first-order destruction rate coefficients), and showing the absolute instead of the relative difference?

That's correct about the relative differences in the stratosphere - a lot of the values in the stratosphere tend to infinity and it misguides the reader to think there are very large (and significant) differences occurring here. Given that there is very little CH4 and H2 in the stratosphere and we're focusing on surface concentrations and lifetime burdens, we do not think the reader is missing any relevant information by masking out the stratosphere (<1% of the total burden of the atmosphere). We had previously considered the 1/lifetime plot, however

decided units of 1/lifetime is not a meaningful way to present the data as it cannot be compared with others' work easily or interpreted.

29. Figs. 8 and A4: I would ask you to perform a statistical significance test on these differences, as the sharp changes in sign could indicate "noise" as well.

We have now removed this section as with further simulations we find a more compelling explanation. See R1 comment #2

30. Table 2 and its discussion: Given the very similar setup and the generally positive reviews, I suggest to also compare the budget terms to the results by Surawski et al., 2025, https://doi.org/10.5194/egusphere-2025-1559.

We have added in the values from Surawski into Table 2.

31. Table 2 and 3, and/or discussion: Please provide a reason for averaging over 2003--2013 here instead of the previously used period from 2008--2013.

This was a typo - changed to read 2008-2013 in tables 2 and 3.

32. l. 273f: Why would a lower CH4 abundance entail a CH4 lifetime reduction? Is this assuming that less CH4 means more OH?

CH4 feeds back on itself through the saturation of OH (Holmes 2018; https://agupubs.onlinelibrary.wiley.com/doi/10.1002/2017MS001196). A large/smaller methane burden causes a longer/shorter lifetime. We have clarified this: "Methane lifetime decreases when including methane flux (0.167 years; 2.1%), due to the slightly lower abundance of methane than the LBC condition. This will reduce the feedback on CH4, which is consistent with the findings from Folberth et al. 2022."

33. l. 307ff: This paragraph is very confusing. Please rework, considering the following points:
   a. No need to speculate on the reason for lower emissions - please check your assumption based on the data you used

   Clarified: "(due to reduced biomass burning)"

   b. Different units for soil uptake and hydrogen production
   c. Single numbers called ranges
   d. Logic hard to follow, especially if/how southern hemisphere aspects should explain northern hemisphere differences

e. Please explain the change in sign of the interhemispheric difference between PI and PD and the reason for the seasonality in the southern hemisphere at PD which is absent in the PI simulation.

We have rewritten the paragraph to address the comments above:

"In the northern hemisphere, hydrogen concentration peaks around April-May in PD run, while in the PI simulation, hydrogen peaks earlier in the year (March).

This is likely due to a smaller chemical source in the summer in the TS-H2CH4-PI simulation, where production only reaches up to 0.38 mol s-1 compared to 0.57 mol s-1 in the TS-H2CH4-PD simulation. The smaller chemical production results in a shift in the peak of surface H2 mixing ratio to earlier in the year, relative to the present day.

Further to this, there is a relative decrease in southern hemisphere emissions in the PI simulation (due to reduced biomass burning) compared to the PD simulation, resulting in a smaller difference in hydrogen mixing ratios between hemispheres when comparing between PI and PD scenarios. Overall, there is a lower seasonal amplitude in the southern hemisphere in the PI scenario due to a smaller seasonal amplitude in the hydrogen chemical production (PI: 0.06 mol s-1 with PD: 0.26 mol s-1), as well as there being no anthropogenic emissions in the PI scenario."

34. Section 4.5 and Fig. 10: This section requires some extension in my opinion.
    a. Please explain why you chose to set a constant mixing ratio throughout the atmosphere instead of actually adding an emission pulse (or use a constant relative increase).
    b. Please explain at least briefly the math and assumptions behind the formulas you provide. Please also rewrite them so that you don't have to take logarithms of units. Furthermore, please comment on why you consider 6 years to be a long enough period for determination of the feedback factor, given the interaction with CH4 that has a perturbation lifetime of the order of 10 years.
    c. I suggest to refrain from citing the preprint by Skeie et al., 2024, https://doi.org/10.5194/egusphere-2024-3079, as the final revised version has already been available in ACP since May 2025: https://doi.org/10.5194/acp-25-4929-2025. Note also that they do no longer report their findings as feedback factors (see discussion during peer review).

    We have updated the reference for Skeie et al. (2024) (see below for full response)

d. Please consider showing the temporal development of the hydrogen burdens instead of the surface mixing ratios in Fig. 10, as they would relate more directly to the feedback analysis.

Due to the set up of the pulse experiment, we realised the pulse simulation would not allow for an analogous calculation of perturbation lifetimes comparable with literature. We have decided to remove this section from the paper as it may otherwise cause confusion and does not directly relate to the H2-CH4 interaction.

**Technical corrections**

Simulation naming: Please either add "simulation" where the abbreviations are used, or use a different naming scheme that identifies the simulation descriptors more easily, e.g., TR-/TS-... for the transient and time slice simulations, respectively. Capital letters would help in general. You might also consider avoiding the semicolon in the identifiers, as it interrupts the reading flow in some places, where it is not immediately clear that you refer to a simulation abbreviation.

Thank you for the suggestion. We've renamed all the abbreviations for better readability

l. 37: Please add a couple of words on the terms "nudged" and "ERA5 data", and a reference for the ERA5 data.

We have added in a reference for the ERA5 interim and the ERA5 online service. We advise the reader to Telford et al. (2008) paper for further information on the setup.

l. 39 (and throughout): I recommend to add the word "fixed" wherever the LBC is mentioned, as a flux is also a lower boundary condition.

Done.

l. 39: which -> with [?] Done.

l. 41: four -> five [according to Table 1]

We have clarified in the text: "A total of four nudged simulations have been performed to compare the interactive fluxes;"

ll. 41ff: This paragraph forces the reader to jump back and forth between text and Table. Please add essential information to the text, e.g., that a "Sanderson" simulation was run in addition to the 4 mentioned transients. It would also be good to mention explicitly that the time slice simulations have free running meteorology. Furthermore, I suggest to add

some motivation for the individual simulations (e.g., Sanderson as standard UKESM hydrogen soil sink parametrization), to make the Section easier to follow.

We have added the following: "In addition to the four simulations, a fifth simulation using the hydrogen deposition scheme from Sanderson et al. (2003) with fixed methane LBC was run (named Sanderson in Table 1) to compare to another deposition scheme."

l. 51, and throughout: concentration -> mixing ratio (where appropriate); and why "number density" for OH, for which you actually do report concentrations?

We have corrected the miss use of concentration when we refer to mixing ratios. However, number density (number per volume) and concentration are the same thing and used interchangeably in the atmospheric chemistry modelling community so we will leave this as it is.

Table 1: I find the different wordings for the same things (if my understanding is correct) confusing.

> "Nudged from 1982." for "Control" means the same as "Nudged from 1982 -- 2013" for the following ones!?

> "LBC for both H2 and CH4. Nudged 1982-2013."

> "CH4 biogenic, biomass burning and anthropogenic emissions + CH4 flux adjustment" for "CH4-flux" means the same as "all CH4 emissions" for "H2;CH4-flux"!?

> For H2;CH4-flux: "Nudged from 1982-2013 with H2 emissions and CH4 biogenic, biomass burning and anthropogenic emissions + CH4 flux adjustment"

l. 58: are -> is Done.

l. 66: Please specify the version of the CEDS data you used. Done

l. 85 (and throughout): I recommend to replace "modelled" by "simulated" where you refer to the output of the simulation, as I would consider "modeling" to refer to the method rather than the output of a model.

We have changed this when describing species (e.g. simulated hydrogen) but have left it when describing the data (e.g. modelled data) as this is a common phrase used in atmospheric modelling and avoid confusion in certain contexts.

ll. 91f: Something went wrong here during editing, I suppose (until "at" in "at 0.0358"). Done

l. 99: parametisation -> parameterisation Done

l. 100: lead a -> lead to a Done

Fig. 1 caption: hydrogen uptake -> deposition velocity of hydrogen uptake

l. 102: Please add a note on the tuning of the soil sink here (cf. ll. 383ff).

Done. "The scheme already has a tuning parameter, alpha, which was scaled to the tropospheric hydrogen global burden to approximately 155Tg to be in line results from Ehhalt & Rohror 2009."

l. 103: Prior paragraph should mention that it describes the soil resistance in a resistor model.

We have clarified in line 96 that we are describing soil resistance: "The model uses a two-layer hydrogen soil uptake scheme to calculate soil resistance (rc) …"

l. 104: hydrogen uptake -> deposition velocity of hydrogen uptake

ll. 105f, and many places: deposition -> deposition velocity Done

l. 109f: The two features should be mentioned in opposite order (or the text below rearranged). It is confusing that the 2nd feature is described first, and the first described in a separate paragraph below. Done

l. 111: Either pull in equations and definitions here, or describe more abstractly (with reference(s) to the Appendix. The formulation as it is requires the reader to jump between this Section and the Appendix, which should be avoided. Done. We have changed the description and referred the reader to the Appendix.

l. 115: Maybe replace "of high volumetric SMC" by "where SMC is of comparable magnitude to soil porosity", as you only refer to the relative value throughout the paragraph.

We have changed it to "At locations of high volumetric SMC relative to the soil porosity, …"

l. 118: This -> The Done

Fig. 2 caption:

      from Paulot -> adapted from Paulot Done

      remove redundant parenthesis "(H2-Flux)" Done

      monthly mean hydrogen deposition -> monthly mean hydrogen deposition velocity Done

l. 128: land use type -> land type (?) Corrected to land-use type

Fig. 3:

      Vertical axis label should be "H2 deposition velocity / cm s-1". Done

      Suggest to add N and E (or W, without minus sign) to the coordinates. We appreciate the detail, but have decided to leave the coordinates as is

l. 136: solid -> dash-dotted Done

l. 144: Gir -> Gif Done

ll. 147f: hydrogen velocities -> hydrogen deposition velocities Done

l. 149: deposition -> deposition velocities Done

l. 152:

hydrogen values -> hydrogen deposition velocity values Done

Since Fig. 4 rather suggests an overestimation of mixing ratios at Mace Head during the period evaluated for Fig. 3, I assume "concentrations" to be a typo here. If so, the clause is redundant, as "underpredicted" already says that the simulated values are below the observed ones.

We have rephrased this sentence: "... and are outside of the range of observed deposition velocities, although within the standard deviation of observations for the remaining months (June to October)."

l. 153: error -> standard deviation of the observations Done

l. 175: 80? From the figure, I read more than 100 ppbv of deviation. Done

l. 185: show the -> show that the Done

ll. 187f: are able to capture ... -> indicate that ... is captured Done

l. 188f: I can only find one "excluded" point, below the colorbar. Are there more? I suggest to move it/them below the Taylor diagram, and would ask you to move the explanation to the Figure caption.

Done and we've corrected the formatting to show all outliers.

l. 195: observed deposition -> deposition comparison Done

l. 200: between -> averaged from Done

l. 201: the magnitude of the hydrogen concentrations -> observed hydrogen mixing ratios to within ... ppbv Done

ll. 210ff: Although you mention sensitivity to chemical fluctuations, and find a 10% change in [H2], you expect minimal impact on O3 and OH!? Please consider rephrasing this paragraph.

We would expect a 10% change in H2 to have a minimal impact on ozone and OH, particularly as soil uptake rather than OH is the main H2 sink. For example, Figure 1 in Warwick et al. (2023) - https://acp.copernicus.org/articles/23/13451/2023/acp-23-13451-2023.pdf - indicates that a 10% change in H2 would only have a minor effect on O3 and OH. We have added the Warwick et al (2023) reference to support the statement.

l. 212: H2-flux -> in the H2-flux Done

Fig. 5 caption:

south -> southern hemisphere Done

north -> northern hemisphere Done

Label correspond is -> Labels correspond to Done

and are given -> as given Done

Fig.s 6, 8, A3, A4: It would be nice to unify the simulation descriptions in the titles (column headers). Done

Fig. 6 and ll. 244f (and analogously Figs. 8, A3, and A4): Please use blue for negative, and red for positive values, and plot "Flux - Control / Control" instead of its negative. The figure will then remain the same, but can be much more easily interpreted as the differences introduced by switching from fixed LBCs to fluxes. Furthermore, I suggest to leave out the middle column, as it does not provide any added value.

We have replotted as (flux - control) / control as suggested for all altitude vs latitude plots, although we decided to keep the middle column in.

Fig. 7: Vertical axes labels should be "Surface CH4" and "Surface H2". Changed so y-axis are matching

l. 241: fluxes -> loss rates Done

l. 242f: You divide by species mixing ratios, and the reactions do not become independent of the species. Please use precise language. What you describe here might be called (aggregate) first-order rate coefficients, or you could directly refer to their inverses, namely the chemical lifetimes. Done

Fig. 8 (3 times): H2 atm -> H2 chem Done

l. 273: with -> which Done

Table 3 caption: 10 years between 2003--2013 to 3sf -> 11 years from 2003--2013 (?) See comment #32

l. 291ff: Please delete the 10 ppb approximation, if the means are 765 ppb vs. 761 ppb, as you write two sentences later. Done

Section A1: Please state explicitly here that you are reproducing the work by Ehhalt and Rohrer, 2013, https://doi.org/10.3402/tellusb.v65i0.19904 Done

l. 307: This -> There [?] Done

Fig. 9 caption: This currently says that both the PI and the PD simulation provide data for 1850. For more precise language and easier reading, you could write "Five-year hemispherical averages of surface hydrogen (blue) and methane (red) mixing ratios for a) 1850 (H2;CH4-PI simulation) and b) 2020 (H2;CH4-PD simulation)." Done

Fig. 10: I suggest to remove the "(2020)" from the title Done

Fig. 10 caption: and -> over Done

l. 348: simulation -> simulate Done

l. 356: to -> of Done

l. 357: THe -> The Done

l. 360: (UM-UKCA).The -> (UM-UKCA). The Done

l. 369: deposition -> deposition velocity Done

l. 376: m3 air / m3 total pore space -> m3 air / m3 soil We have kept this as is for clarity (different schemes define "soil" differently)

l. 376: m3 total pore space / m3 total volume -> m3 total pore space / m3 soil See comment above

l. 378f: I suggest to use SI units here. We have used the equation as written from Ehhalt & Rohror as left it as they (and Paulot et al) defined it

l. 378ff: Please use either T or $T_s$ throughout, when referring to soil temperature. Done

l. 379: C° -> °C Done

l. 380 (analogously l. 388, l. 393, and l. 394): $f(\Theta_a)$ -> $f(\Theta_w)$ [and $\Theta_w$ needs to be defined] We have clarified what theta_a is in line 398 (where theta_w is also defined)

l. 386: Units are missing in the exponentials. There are no units for the values in the exponentials aside from temperature (given in 379)

l. 391f (analogously l. 395f): look like a mistake in the reproduction of the work by Ehhalt and Rohrer, 2013, https://doi.org/10.3402/tellusb.v65i0.19904, where limits of applicability are given for the equations in terms of allowed ranges of $\Theta_w$/$\Theta_p$ ratios We have corrected this typo

l. 392: $\Theta_a$ -> $f(\Theta_w)$ Please see 3 comments above

Fig. A2: Please clarify the "emission" label. Done

Fig. A2 caption: Delete closing parenthesis. Done

Fig. A3 caption: All scales are actually linear. Done

Fig. A4:

    LCB -> LBC (two instances) Done

    It looks as if the values of the differences actually span a much narrower range than the colorbar - could be adjusted. We have decided to leave this as is

l. 397ff: If NLA "provided the model runs for all simulations", what is meant with "MAJB ran the simulations"? Remove "runs" for clarity

l. 399: contirubted -> contributed Done

l. 400: provivded -> provided Done

l. 448: delete one instance of "https:doi.org/" Done

l. 533: I suggest to cite the GRL article (https://doi.org/10.1029/2024GL112445) instead. Done

**Reviewer 2 Comments**

Review of "Development of Fully Interactive Hydrogen with Methane in UKESM1.0" by Brown et al for publication in GMD

The manuscript documents a more comprehensive configuration of UKESM1 that includes the simulation of hydrogen and methane driven by emissions rather than prescribed lower boundary conditions (LBCs) to capture the complex interactions and feedbacks between these species as well as impacts on atmospheric composition. The authors perform a number of sensitivity simulations to assess the effects of replacing LBCs with emissions of hydrogen and methane. The model configuration driven by emissions of both hydrogen and methane is generally able to capture the observations, though there are some peculiarities that need deeper assessment (more below). Specifically, the authors find a reduction in global mean surface methane concentration (and a drift) in this configuration characterized by higher global mean surface hydrogen concentration and the explanation given for this is not convincing.

The authors apply the emissions driven configuration to simulate preindustrial concentrations of hydrogen and methane and estimate the hydrogen feedback factor to demonstrate the capability of this model configuration.

The manuscript has some shortcomings which can be overcome with better organization of the material, and improved and additional analysis to support conclusions. The description of the model configuration falls within the scope of GMD and is novel enough to warrant publication. I encourage the publication of this manuscript after my comments below have been addressed.

We'd like to thank the reviewer for their time and consideration in these helpful replies. We have made several large changes to manuscript, including running some new simulations (which are outlined in R1 comment #2) and addressed all of their comments below.

**Specific Comments:**

1. L17-18: While the main "chemical sink" of hydrogen is oxidation by OH, but it accounts for less than one third of the total H2 sink (Ehhalt and Rohrer, 2009; Paulot et al., 2021). It would be good to note that.

   We have now addressed this in #3

2. L21-23: "Earth system models (ESMs) currently do not account for both hydrogen and methane fluxes at the surface,..." why? It would help to have an answer for this question as a justification of why this effort is undertaken here.

   We have added the text below on line 21 which outlines the reasons for using methane and hydrogen LBCs and the benefit of replacing these with flux schemes:

"Earth system models have typically prescribed methane and hydrogen as fixed lower-boundary concentrations because their sources and sinks are uncertain and their long lifetimes make them sufficiently well-mixed to reproduce global burdens without explicit fluxes. This approach also helps prevent model drift that can arise from poorly constrained fluxes. Incorporating surface fluxes, however, enables models to better represent chemistry-climate feedbacks and thus generate more realistic projections of future atmospheric burdens and radiative forcing."

L26: elaborate on "methane feedback factor"

See comment R1 #4. We have also added in a reference to (Holmes 2018)

3. L27: why is a deposition scheme important? This is where the soil sink of H2 comes into a picture.

   We have adapted the sentence in line 17: "The main sink for atmospheric hydrogen is by soil uptake (~70%), while the chemical sink via reaction with the hydroxyl radical (OH) accounts for 30% Ehhalt & Rohror 2009."

4. L34-40: an amip configuration is not the same as nudged. What is the motivation for `choosing the option with Hydrogen time-invariant. And why are the LBCs uniform spatially?

   Spatially varying LBCs were not available until after these simulations were run (see Bryant & Stevenson 2024: https://rmets.onlinelibrary.wiley.com/doi/10.1002/wea.4567). The global hydrogen mixing ratio has not changed significantly between 1982 and 2013 (increase of 500 ppbv to 550ppbv) and as H2 observations are sparse for pre-2000s, so we felt it was not necessary to include a time varying H2 LBC.

5. L43-44: "As hydrogen deposition has a very limited interannual variability,"- please provide supporting reference for this statement and elaborate how this compares with the findings of Paulot et al (2021) and Derwent et al. (2021).

   We have addressed this in R1 comment #6

6. L45: what meteorology is used to drive the timeslice simulations? Are these driven by climatological SSTs/sea-ice or are they nudged to repeated winds/temperature for a specific year?

   We have added the following text to explain timeslice simulations: "A timeslice experiment involves running the model with fixed sea surface temperatures, sea ice, and, all other boundary conditions for a given year."

7. L47-50: Anthropogenic emissions of H2 are not available for CMIP6 historical and scenarios based on data holdings in input4MIPS

We had rephrased the sentence: "...with the former using CH4 and H2 (derived from CO) emissions from the Coupled Model Intercomparison Phase 6 project, using the Shared Socioeconomic Pathway 3 with a radiative forcing of 7.0 Wm-2 at the end of the 21st century i.e., SSP3-7.0 (see Section 1.2 for the calculation of H2 emissions)."

8. https://aims2.llnl.gov/search?project=input4MIPs. Rather than a blanket statement, the authors should point the readers to the detailed description of emissions in section 2.2. Please elaborate on the reasoning behind a 30 year spin up.

We have addressed this in comment #7 and R1 comment #4.

9. L50-53:
   a. these sentences can be combined

   Done

   b. what does at all levels mean - all vertical levels? If so, why was this done throughout the vertical column and not at the surface only

   We have addressed this in R1 comment #34. And clarified in line 54:

   "...experiment was conducted by setting the initial hydrogen concentration at all levels (vertical and spatial) for the first timestep to 25% higher than the global-averaged surface concentration in the TS-H2CH4-PD simulation (from 530 ppbv to 662.5 ppbv). The same starting conditions as the TS-H2CH4-PD simulation after the spin up."

   c. was the hydrogen concentration set to 45.7ppb or 662.5ppbv? How long was the pulse (increased H2 concentration) implemented for - the full duration of the simulation or just one year? If it was implemented for the full duration of the simulation, then this is technically not a pulse simulation but a sustained perturbation.

   45.7ppb or 662.5ppbv are the same values given mass mixing ratio and volume mixing ratio respectively (we use the mass mixing ratio as the input for the model, but included the volume mixing ratio in the text as it is the more commonly used unit). To avoid confusion we have removed the mass mixing ratio value from the text. Also see comment above.

10. Table 1: The experiments need better naming. The H2;CH4-X construct is confusing. I would suggest something like H2_CH4-X, but I am sure the authors can be more creative.

    We have renamed the schemes - see first technical comment of R1

11. L55-78: The description of emissions of H2 used in the simulations can be improved. A few points to consider -
    a. the BB4CMIP (van marle et al) emissions provided H2 biomass burning emissions explicitly, and it sounds like oceanic and terrestrial H2 emissions are also used from available sources, so this statement "The ratio of CO:H2 for each category (anthropogenic, biomass burning, oceanic and terrestrial) is derived from the period 1995−2014, where we have a known estimate of hydrogen emission" needs to be clarified. Further, "The resultant hydrogen emission for each source (anthropogenic, biomass burning, oceanic, and terrestrial) follows the spatial pattern of the equivalent CO source, but with values rescaled to give the global emission total appropriate for hydrogen" should also be clarified.

    We have now addressed this in R1 comment #8

    b. how do the emission totals compare with those from Paulot et al (2021)? This comparison should be noted in the text.

    We have also address this in R1 #8

    c. what are the anthro and bb total emissions for years 2020 and 1850? These are used in simulations, but not discussed/displayed anywhere in the manuscript.

    We have now included further information on the H2 emissions for the PD and PI simulations in line 76:

    "The total biomass burning and anthropogenic H2 emissions accounted for 7. Tg yr-1 and 14.9 Tg yr-1 respectively. The TS-H2CH4-PI simulation used the same biomass burning hydrogen emissions as calculated for the nudged simulations, which extend back to 1850 and contributed 13.1 Tg yr-1."

12. L81-82: How are biogenic emissions of methane implemented? Are wetland emissions prescribed or interactive? If prescribed, please elaborate on the source. I am sure all of this information is included by Folberth et al (2022), but I hope the authors agree that it would be cumbersome for the readers to go digging into that paper to understand the results here.

    Biogenic emissions of methane are not treated separately to other emissions sources. We have added a sentence to clarify wetlands: "Biogenic wetland emissions and soil uptake are calculated interactively within the model, whereas all other emissions are prescribed."

13. L87: clarify what "Note that wetlands are excluded" means. I assume you mean that the flux adjustment is not applied to wetland emissions because they are calculated interactively.

We have updated Section 2.3 with the text below to clarify what the flux adjustment represents, and how the flux adjustment varies between the different simulations:

"Methane emissions used in the CH4 flux experiments include biomass burning, anthropogenic, and biogenic emissions. Biogenic wetland emissions and soil uptake are calculated interactively within the model, whereas all other emissions are prescribed (Hoesly et al. 2018; Fung et al. 1991).

To reconcile simulated global mean surface methane concentrations with observations, a residual methane surface exchange flux, or flux adjustment, is applied in all simulations, following Folberth et al. (2022). This flux represents the net "missing" sources or sinks required in the emissions-driven configuration of UKESM to capture global methane observations.

Separate flux adjustments are used for the pre-industrial (TS-H2CH4-PI) and present-day (TS-H2CH4-PD) timeslice simulations, as well as for the nudged simulations (TR-CH4 and TR-H2CH4-flux). The PI and PD adjustments are taken directly from Folberth et al. (2022), with magnitudes of ~5 Tg yr$^{-1}$ and 48 Tg yr$^{-1}$ respectively. In the nudged simulations, a larger adjustment is required than in the PD timeslice because interactive wetland emissions are lower (135 Tg yr$^{-1}$ versus 190 Tg yr$^{-1}$ in the PD simulation). This reduction in wetland emissions is consistent with a smaller global wetland fraction in the nudged configuration (0.0321) compared with the PD timeslice simulation (0.0358, an 11% increase). To compensate for the reduced wetland source, the flux adjustment from the PD simulation was scaled by 3.5 in the nudged simulations to enable the model to capture the growth in surface global mean methane during the 1980s. Note that the flux adjustment is static across all years.""

14. L88-90: How can the authors tell that the underestimate in methane is due to an underestimate of wetland emissions or for that matter other source emissions or an overestimate of the methane sink (hydroxyl radical)? I think there is an assumption being made that emissions from all other sources are well-constrained and so is the modelled hydroxyl radical. If so, the authors should provide a basis for this assumption.

Please see the reply to the comment above. The flux adjustment represents the residual methane exchange needed to align modelled methane mixing ratios with observations, and therefore reflects the net missing sources or sinks in the system. We do not interpret the model's methane underestimate as evidence that wetland emissions are intrinsically too low; rather, the reduced wetland emissions in the nudged simulations simply mean that a larger residual flux is required to achieve agreement with observed methane concentrations.This has now been clarified in Section 2.3.

15. L104: by "global hydrogen uptake", do you mean the global mean hydrogen deposition velocity (especially since the units are cm s-1)?

We have changed the caption to read deposition velocity

16. L106-107: Figure 1 shows deposition velocity. If you meant to show hydrogen uptake, please update the figure. Also, what year are these deposition velocities for? Do they vary between the transient and timeslice simulations?

   In line 107 we have added "averaged between 1982-2013." and changed the instances of uptake to deposition velocity.

17. L132: It would be prudent to be precise in the text here and throughout the manuscript. Figure 2 shows the comparison of H2 deposition velocity calculated based on two two deposition schemes.

   We have rephrased the sentence to: "Figure 2 shows the comparison of hydrogen deposition velocities between the scheme adapted fromPaulot et al. (2021) and the scheme from Sanderson et al. (2003)."

18. L149: I am confused, did CMIP6 provide H2 deposition velocities?

   No, the deposition velocities are not given in CMIP6 data. We have rewritten line 149 to clarify: "Shaded areas show the range of deposition velocities from CMIP model inputs calculated in Brown et al. (2025)."

19. L162: "verify the integrity of the model" sounds strange. How about "evaluate the skill of the model?"

   Agreed. We've made this change

20. Figure 4 caption: I am assuming the years in the parentheses in "hydrogen (1994-2013) and b) methane (1985-2013)." indicate the years of observations and not the simulations. Please revise the caption to be precise.

   We have indicated this corresponds to the observations

21. L164-165: What is the source of these observations? It would be appropriate to give credit to those who make these observations by providing a reference and/or doi for the data. I would also recommend adding a separate Observations subsection under the Methods section to describe them and their associated uncertainties, as well as the reasons for choosing specific sites (e.g., longer timeseries, remote station capturing background concentrations etc).

   We have added the correct reference to these data. We have also added in a sentence to explain why these sites were chosen on line 166: "These sites were chosen as they had the longest continuous H2 record available in the Northern and Southern Hemispheres, as well as their locations' suitability for monitoring baseline atmospheric conditions."

22. L180-181: While the CMIP6 historical emissions ended in 2014, the timeseries could be extended using an SSP scenario. The CMIP7 historical emissions are now available but of course the model simulations cannot be rerun with these emissions.

Indeed as the reviewer points out, these would not be accessible nor feasible as the simulations would have to be rerun.

23. Figure 4: please replace "model" in the labels with the title of the simulation being shown for better readability.

Done

24. L184-185: Do all sites provide data for the same time period? If not, I would suggest adding a column to Table A1 to indicate the years of data available for each site.

Done

25. L200: "averaged hydrogen concentration from all sites given as points", this comes across as the average of data from all sites. Please change "all sites" to each site.

Done

26. L202-203: How does orography contribute to high simulated H2 concentrations over India and China? Are high values also simulated for CH4 over these regions because of orography?

We have addressed this in comment #20 of Reviewer 1's comments

27. L209: "Effect on Atmospheric Composition" of what?

This refers to the addition of H2 and CH4 flux (as implemented also in the previous section heading). We have changed the heading to "Impact of Fluxes on Atmospheric Composition"

28. L210-212: Please point to the figure upon which this conclusion is based.

We have added in a reference (the figure is mentioned in the next sentence).

29. L213-216: Since the discussion is focused on the effects of interactive (emission-driven) CH4 and H2 simulations on atmospheric composition, it would be more logical to subtract the control from the Flux to calculate the percent change [(Flux-control)/control].

We have replotted all relevant figures with this method

30. L218-219: I see the peak in 1991 in BB emissions in figure A2. 1992 is also the year of the Pinatubo eruption. Can there be any effects on H2 from that eruption?

We do not expect to see any impact from the Pinatubo eruption on methane or hydrogen in Figure 7 as volcanic emissions are not included in our simulations. We are aware that aerosols emitted from the Pinatubo eruption caused a large depletion in stratospheric ozone, which will have allowed more UV through to the troposphere resulting in an increase in OH and therefore a decrease in the observed methane growth rate over that period. However, given that the main sink of H2 is soil uptake

rather than OH, we anticipate any impact on H2 via changes in OH would be small. H2 is also emitted in small amounts from volcanic eruptions, but again we anticipate any impact on global H2 mixing ratios would be minor.

31. L220: "The addition of the methane flux" - this should technically be "replacement of the methane LBC to flux" since you are not adding a new emission flux to the simulation, rather replacing the CH4 LBC with emissions.

    Done

32. L220: The effect of methane is included in both the H2-flux and H2; CH4-flux simulations. It is only how the effect of methane is included that is different. Please ensure this is corrected throughout the manuscript. Further, because of the adjustment applied to the CH4-flux simulations, the expectation is that methane in both the emissions-driven and prescribed simulations will be similar with small differences (~30 ppb, as demonstrated in Fig 7b). Hence, the impact on hydrogen is simulated to be small going from the H2-flux to the H2; CH4-flux simulation.

    We have further added a section (after line 262) from the additional simulations we have done to show the response of CH4 to H2. (See R1 comment #2)

33. L223-224: Why is there a cross over in 1992 in CH4 concentrations simulated for the emission driven run versus prescribed CH4 in Figure 7? This was also evident from Fig 4b for Mace Head. Also, why is there a drift in the CH4 concentrations for the H2;CH4 Flux simulation?

    Methane mixing ratios in the CH4-flux driven simulations do not exactly follow the trend prescribed in the LBC simulation, which is based on observations. This discrepancy is expected given the uncertainties in methane emission inventories. In the nudged CH4-flux simulation, we apply the flux adjustment to bring modelled methane mixing ratios broadly into agreement with observations, without attributing this residual to any particular source or sink - a task which would require a dedicated methane-focused study.

    This flux adjustment is constant over time, so it cannot correct differences in the methane trend between observations and that simulated by UKESM using prescribed emission inventories and interactive wetland emissions. The cross over in 1992 is an artifact of the chosen adjustment; a different value would shift this point accordingly. However, we acknowledge that the CH4 flux model set-up does appear to do a better job of capturing the trend between 1983 and 1992, than 1992 to 2007. Although a time-varying flux adjustment could force the CH4-flux simulation to more closely follow the observed trend, doing so would require substantial additional effort for limited benefit and would not illuminate the underlying causes of the mismatch between modelled and observed methane - which is not the subject of this model development study.

34. The difference between simulated methane concentrations for CH4 Flux and H2;CH4 Flux simulation increases progressively from 1983 (beginning of the simulation) to

2013 (end of simulation). Additionally, there is a gap in 1986 for H2;CH4 Flux which needs to be addressed.

We have addressed the divergence in CH4 in comment #25 of Reviewer 1, as well as the plot.

35. L225: What is meant by good agreement? Is 30 ppb difference (noted in the previous sentence) between the CH4 emission driven runs and the control considered good agreement?

A 30 ppbv difference for a global averaged surface mixing ratio of CH4 corresponds to a ~1.7% difference which we would say is in good agreement. We have added the % difference for clarity.

36. L239-240: This is confusing -"Figure A3 shows the hydrogen and methane chemical loss; H2 chemical loss increases when H2 flux is implemented into the model, while the CH4 loss via OH decreases", while the figure itself shows CH4-flux minus H2;CH4-flux. Please make it easier for the reader to understand by showing H2;CH4-flux minus CH4-flux which would explicitly show the effect of implementing H2 flux and be consistent with what is written in the text. If CH4 loss via OH decreases when implementing H2 flux, then why does the methane lifetime stay the same between CH4-flux and H2; CH4-flux simulations (Table 3 last two rows)?

We have now addressed this in R1 comment #2.

37. L242: What is meant by "the OH activity cannot be identified."

We have rephrased the paragraph now (see R1 comment #2)

38. L244-245: "Blue (red) indicates an increase (decrease) when H2 flux is included, relative to the simulation with H2 LBC" - this is not consistent with what is actually being shown in the figure. Please revise to show 100* (H2; CH4-flux minus CH4-flux)/ CH4-flux.

Thank you for pointing this out. We have changed the text and revised the plot.

39. Figure 8: Please include the full atmospheric column in the figures, particularly because you are using the full atmospheric burden and loss rates to calculate the lifetimes. Masking out just because of strong variability in the stratosphere is not appropriate. Further, please assess the significance of the differences being shown. Would you consider % differences of the order of 0.1% significant?

We have addressed this in Reviewer 1's comments #28 and #29

40. Table 2: Please show the H2 budget terms for all the simulations conducted for completeness and ease of comparison. According to mass balance, the sum of all production (atm prod + emissions) terms should be equal to the sum of all loss (soil + atmospheric) terms at equilibrium. This is not the case for any of the simulations being shown here (the imbalance ranges from 2.3 to 17.8 Tg). Can you please explain this, especially the mass imbalance for the PI simulation?

We don't show the budget terms for simulations which have a fixed H2 LBC as we don't have sources/sinks and cannot calculate the lifetime. We have not included the pulse experiment here as this is discussed later (with the calculation of the perturbation lifetime).

We realised there was a typo in the PI budget for atmospheric production, which we have now fixed (16 Tg yr-1 → 32 Tg yr-1)

41. Table 3: Budget terms for CH4 (prod, loss, deposition), should also be included which will help shed light on the effects of coupling of both emission-driven hydrogen and methane on methane lifetime.

Methane is not chemically produced in the atmosphere. The methane lifetimes reported here already reflect the variation in methane loss due to reaction with OH across simulations. In the LBC simulations, methane is prescribed as a fixed concentration rather than a surface flux, so soil deposition is not represented. In the CH4 flux simulations, soil uptake (~31 Tg yr$^{-1}$; Folberth et al., 2022) is included, but it is small relative to loss via OH. We therefore do not consider it necessary to include methane deposition in the table for the CH4 flux simulations, as it would not provide additional relevant information.

Tables 2 and 3: please provide a motivation for averaging over 2003-2013 which is different from the averaging period for the figures.

This was a typo and has since been corrected (see Reviewer 1 comment #32)

42. L267-268:Thus far the PI and PD simulations have not been discussed. I don't think these results should be included in the range of lifetime estimates in the text without providing a context. Rather the focus here should be on explaining the differences in budget terms related to the emissions-driven versus prescribed experimental setup.

We have added in the following to make this clear to the reader in line 266: "Budget for the PI and PD timeslices are discussed in Section 4.4." Budget terms for the PI and PD simulations are now also referenced in line 286.

43. L279-280: Based on the results shown here I am not fully convinced that coupling of both interactive hydrogen and methane may cause a decrease in the methane lifetime. I agree with the next statement that further work is needed and perhaps the authors can do a better job at presenting the analysis here. The authors may want to consider running the CH4-flux simulation with H2 LBC increased by, say 10%, to roughly mimic the increased H2 concentration in the H2; CH4-flux simulation. This could help isolate the impact of increased H2 on methane abundance.

We have run some additional simulations which we have discussed throughout the manuscript. Please see R1 comment #2.

44. L283: Please remind the readers, the motivation for analyzing the PI and PD simulations.

We have added the following: "One application of the combined CH4-H2 emissions driven capability is to explore drivers of the combined CH4-H2 evolution over the industrial period. The links between the two species provide a new opportunity for joint constraints on uncertainty in the emissions and sinks of these two gases."

45. L284: The hydrogen burden of 129.4Tg in PI simulation is inconsistent with that presented in Table 2 (136 Tg).

Thank you for pointing this out. We've double checked the burden and corrected it in the text.

46. L286-287: How is the conclusion derived?

We have revised this and now removed this sentence

47. L291: replace south with the southern hemisphere.

Done.

48. L288-294: Any thoughts on why the PI simulation is not able to capture the CH4 N-S gradient derived from ice cores?

The pre-industrial methane budget remains poorly constrained, and the small discrepancy between the modelled latitudinal gradient and that inferred from ice cores could have multiple causes. Investigating these causes would require additional constraints (e.g. methane isotopologues), which are beyond the scope of this study.

49. L316-326: Is the methodology used here to assess the H2 feedback factor and the perturbation lifetime developed in this study or based on previous studies? No references are provided which gives the impression that this is original work. Please clarify. What is the reason for choosing the 6-year decay time?

We have now removed this section. Please see our response at the beginning of the manuscript and R1 comment #36.

50. L339: "which has been tuned to literature values of the tropospheric hydrogen burden" did I miss something? Was there an adjustment applied to the H2 emissions or deposition? I may have lost track.

Yes, there was the alpha parameter for the deposition which was tuned (lines 383). We have clarified now this in lines 102: "The scheme also has a tuning parameter, alpha, which was scaled to the tropospheric hydrogen global burden to approximately 155Tg to be with results from Ehhalt & Rohror 2009." and in line 339: "...which has been tuned via the soil sink (see Appendix A) to literature values of the tropospheric hydrogen burden…"

51. L343: "little change in hydrogen concentration when CH4 flux was added" - please revise to "...when CH4 LBC was **replaced** with CH4 flux".

Done

52. L348: replace simulation with simulate

Done

53. L347-349: But there are biases for PI methane. The statement comes across as overselling the model's skills. Calling a spade a spade will ensure credibility!

Our simulations use methane flux adjustments - residual emissions that effectively account for missing sources or sinks - to align simulated methane mixing ratios with observations. The small discrepancy for the PI period arises because we apply the flux adjustments from Folberth et al. (2022) to a slightly different model configuration, which has a marginally different atmospheric methane lifetime. We acknowledge that this introduces a minor bias in PI methane (~5%), but a detailed analysis of the methane budget is beyond the scope of this study. With a 5% bias in PI methane, we feel that we are not overselling the model's ability when we say 'When H2 flux and CH4 flux are included, the ESM is able to simulate PI conditions which are within a similar order of magnitude as concentrations found in firn measurements.'

54. Figures: There is an inconsistency in the choice of color bases across the figures (e.g., Fig 8 top and bottom in columns 1 and 2 have different colorbars. why?). Please be consistent.

Colourbars are assigned to each chemical species, rather than each plot to clearly show different chemical species.

55. Throughout the manuscript there is inconsistency in the labeling of simulations (e.g., CH4-Flux versus H2 LBC;CH4-Flux). Please ensure consistency in the labeling.

Done